# Dynamics underlie the drug recognition mechanism by the efflux transporter EmrE

Jianping Li[1,3], Ampon Sae Her[1,3], Alida Besch[1,3], Belen Ramirez-Cordero [1], Maureen Crames[1], James R. Banigan [1], Casey Mueller[1], William M. Marsiglia[1], Yingkai Zhang [1,2] & Nathaniel J. Traaseth [1] ✉

The multidrug efflux transporter EmrE from *Escherichia coli* requires anionic residues in the substrate binding pocket for coupling drug transport with the proton motive force. Here, we show how protonation of a single membrane embedded glutamate residue (Glu14) within the homodimer of EmrE modulates the structure and dynamics in an allosteric manner using NMR spectroscopy. The structure of EmrE in the Glu14 protonated state displays a partially occluded conformation that is inaccessible for drug binding by the presence of aromatic residues in the binding pocket. Deprotonation of a single Glu14 residue in one monomer induces an equilibrium shift toward the open state by altering its side chain position and that of a nearby tryptophan residue. This structural change promotes an open conformation that facilitates drug binding through a conformational selection mechanism and increases the binding affinity by approximately 2000-fold. The prevalence of proton-coupled exchange in efflux systems suggests a mechanism that may be shared in other antiporters where acid/base chemistry modulates access of drugs to the substrate binding pocket.

Bacterial antibiotic resistance is an urgent global health problem fueled by the slow pace of antibiotic discovery and the emergence of resistance mechanisms that reduce the effectiveness of drugs[1–11]. Efflux of antibiotics confers a broad resistance mechanism used by pathogenic bacteria to bind and transport drugs across the membrane in a promiscuous manner[12–14]. Efflux transporters rely on proton binding through acidic residues as a way of harnessing the proton motive force (PMF) for drug removal. Structures of efflux transporters have revealed the presence of multiple conformational states in the transport cycle and the location of drug binding within the substrate binding pocket[14,15]. However, elucidating the proton-coupling mechanism has been more challenging since the protonation states of the catalytic anionic residues are not easily resolved in crystallography or cryoelectron microscopy experiments, thereby creating a disconnect between structural and mechanistic studies[16].

NMR spectroscopy is a sensitive method for characterizing structure, conformational dynamics, and protonation states, features essential to explain the transport mechanism of efflux transporters. Due to its amenable size of 110 residues, we used the *Escherichia coli* efflux transporter EmrE as the model system to study ion-coupled transport using NMR spectroscopy. EmrE is the archetype member of the Small Multidrug Resistance (SMR) family of transporters, which are found in bacteria and archaea and form homo- and heterodimers consisting of four transmembrane (TM) domains in each monomer[17–20]. A subset of proteins within the SMR family, like EmrE, are members of the subfamily of DHA4 efflux transporters which antiport drugs by coupling to the PMF. EmrE shares conserved features found in other transporter families, including a membrane embedded acidic residue (Glu14) that is indispensable for drug efflux[21].

EmrE is a dual topology protein where the two monomers in the dimer are oppositely oriented in the inner membrane of *E. coli*[22,23]. Substrates are moved across the membrane through a rocker-switch mechanism, resulting in accessibility of the substrate binding pocket to the cytoplasmic or periplasmic side of the membrane[24–26]. Covalent

[1]Department of Chemistry, New York University, New York, NY, USA. [2]Simons Center for Computational Physical Chemistry, New York University, New York, NY, USA. [3]These authors contributed equally: Jianping Li, Ampon Sae Her, Alida Besch. ✉e-mail: traaseth@nyu.edu

crosslinking of the dimer prevents alternating access and displays a loss of efflux activity[27], further underscoring the functional role of the anti-parallel structure. Recent atomic resolution models harmonize with the overall asymmetric and anti-parallel dimer quaternary structure of the earlier structural work and provide new insight into the transport cycle of EmrE. Namely, solid-state NMR spectroscopy in lipid bilayers was used to reveal how the high-affinity substrate tetraphenylphosphonium (TPP) changed positions in the substrate binding pocket as a function of protonation of Glu14 from one of the monomers[28,29]. These findings offered a structural basis for observations showing that EmrE binds and transports TPP in the singly or doubly deprotonated states of Glu14[30]. Likewise, EmrE crystal structures bound to different substrates revealed intermolecular contacts between EmrE and compounds varying in structure[31]. Notably, the similarity of the proton-bound and TPP-bound structures (0.376 Å backbone r.m.s.d.) suggests drug binding occurs predominantly through the competition model[32], which postulates protons and drugs compete for binding to Glu14 residues in the substrate binding pocket.

Observations from our group found that Glu14 deprotonation within EmrE induces rather large NMR spectral changes[33] that coincide with a ~2000-fold increase in the binding affinity to TPP[34,35] (e.g., Supplementary Fig. 1a–c). However, none of the available structural models explain this correlation, leaving the molecular basis underlying drug binding unclear. In this work, by probing molecular dynamics of essential states in the transport cycle and elucidating NMR structures

of the proton-bound and TPP-bound forms of EmrE, we reveal how Glu14 deprotonation in one monomer of the dimer induces a structural change that enables molecular recognition through a conformational selection mechanism.

## Results

### Drug binding through conformational selection

To probe the molecular recognition mechanism of EmrE, we performed solid-state NMR experiments of EmrE reconstituted in DMPC lipid bilayers using the magic-angle-spinning (MAS) technique. Our experiments focused on probing the side chain chemical shifts of tryptophan residues with $^{15}$N/$^{13}$C correlation spectra due to the central location of Trp63 in the substrate binding pocket (Supplementary Fig. 1d). The NMR spectrum of EmrE at pH 5.0, corresponding to protonation at both Glu14 residues[36], displayed two homogeneous signals for Trp63 monomers A and B at 125.5 ppm and 127.5 ppm in the $^{15}$N dimension, respectively. These two signals indicated a well-ordered conformation and underscored the asymmetry of the homodimer (Fig. 1a, left and Supplementary Fig. 1e)[26]. In contrast, the tryptophan spectrum for EmrE at pH 9.0, corresponding to a deprotonated form, displayed two weak signals for monomer A of Trp63 at 125.1 ppm and 136.9 ppm in the $^{15}$N dimension, only the former of which agreed with the position in the proton-bound conformation (Fig. 1a, middle). The monomer B signal of Trp63 remained relatively homogeneous at 126.7 ppm in the $^{15}$N dimension and at a similar position as in the proton-bound state. Drug binding was investigated by addition of TPP to EmrE

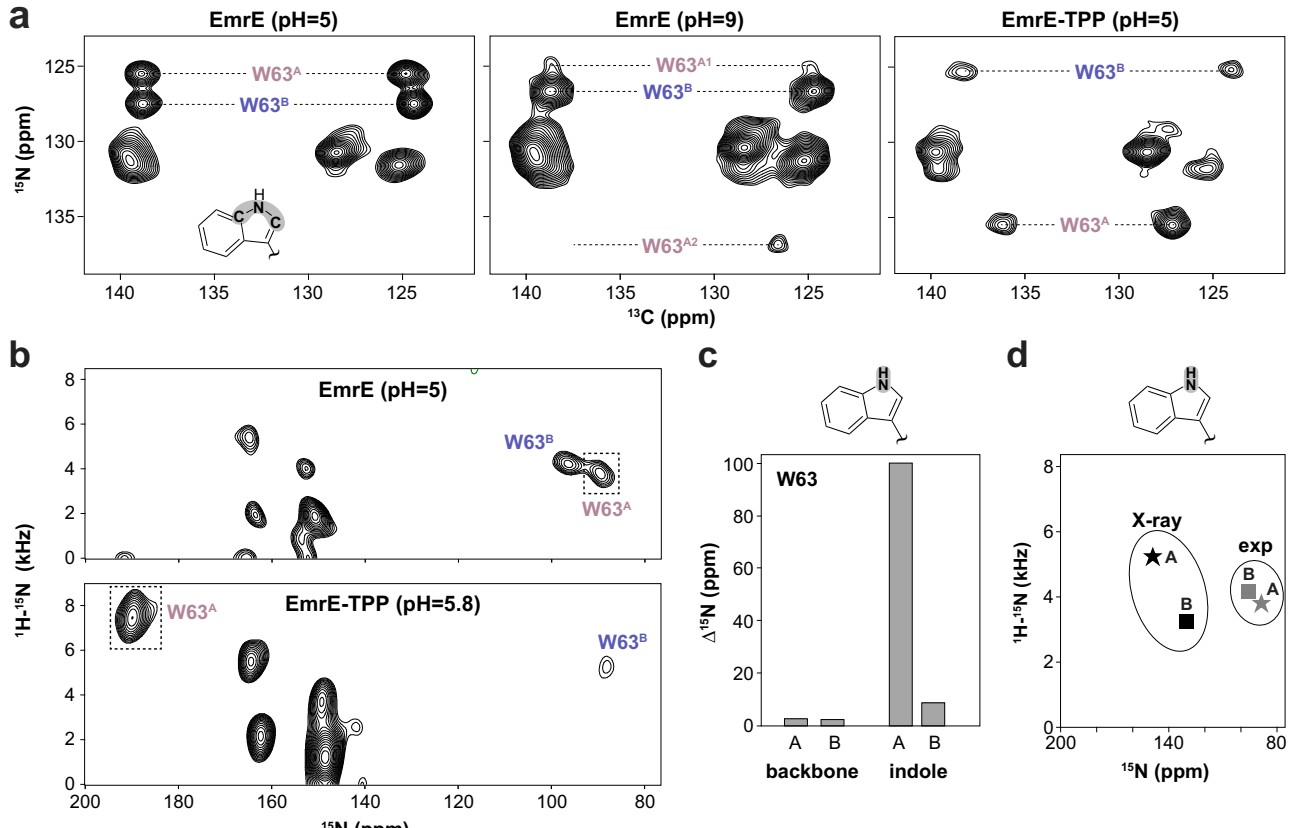

**Fig. 1 | pH induced conformational change of Trp63 in monomer A of EmrE.**
**a** $^{13}$C/$^{15}$N MAS correlation spectra of EmrE in DMPC lipid bilayers probing the highlighted atoms of the tryptophan indole ring in the left panel. Panels correspond to EmrE at pH 5 (left), EmrE at pH 9.0 (middle), and EmrE bound to TPP at pH 5.0 (right). Superscripts "A" or "B" refer to the corresponding monomer A or B, respectively, and "A1" and "A2" denote the two monomer A peaks observed at pH 9.0. **b** $^{1}$H/$^{15}$N PISEMA spectra of $^{15}$N-indole tryptophan labeled EmrE in aligned lipid

bicelles in the proton-bound state at pH 5.0 (top) and in the TPP-bound state at pH 5.8 (bottom). **c** $^{15}$N chemical shift difference plot (absolute value) for tryptophan indole and backbone PISEMA spectra for Trp63 from aligned lipid bicelle samples. **d** Comparison of experimental $^{1}$H/$^{15}$N PISEMA peaks for the Trp63 indole (gray; from **b**) and the calculated Trp63 indole from the X-ray structure (black; PDB ID 7MH6[31]). Stars denote monomer A and squares denote monomer B.

at pH 5, which subsequently revealed homogeneous peaks at 135.5 ppm for monomer A and 125.3 ppm for monomer B (Fig. 1a, right and Supplementary Fig. 1f). Notably, the monomer A chemical shift for the TPP-bound state agreed with the peak position for the second population of monomer A in the deprotonated sample, while the monomer B chemical shift remained similar to that of the drug-free spectrum. Hence, these experiments indicated that deprotonation of Glu14 triggered two conformations in Trp63 of monomer A such that it populated states resembling those of the proton-bound and TPP-bound conformations. Taken together with the improved binding affinity at basic pH values (Supplementary Fig. 1a), these data indicated TPP binding occurred through a conformational selection mechanism, which was induced by an equilibrium shift toward the drug-bound conformation following deprotonation at Glu14.

To complement these experiments, we performed oriented sample solid-state NMR experiments using $^{15}N$ labeling of the side chain indole of tryptophan residues. PISEMA experiments correlate $^1H$-$^{15}N$ dipolar coupling with $^{15}N$ anisotropic chemical shift and are sensitive to relative orientations with respect to the bilayer normal[37,38]. Remarkably, we observed a -100 ppm $^{15}N$ chemical shift change for Trp63 in monomer A between the proton-bound and TPP-bound states (Fig. 1b, c). The chemical shift change for Trp63 of monomer B was relatively minor in comparison and consistent with the observations in MAS experiments. Overall, the large perturbation observed for Trp63 of monomer A was not explained by X-ray crystal structures of EmrE (drug-free at pH 5.2; TPP-bound at pH 7.25), which displayed essentially the same structure bound or unbound to TPP[31]. Specifically, we found that the chemical shifts for Trp63 in PISEMA experiments deviated from the calculated chemical shifts from the purported proton-bound conformation of EmrE (PDB ID: 7MH6) (Fig. 1d). Due to this difference with our experimental data, conditions known to correspond to Glu14 in the protonated state, we rationalized an NMR structure of proton-bound EmrE would be valuable to ensure the conditions used to determine the structure corresponded to a well-defined state in the transport cycle.

## NMR structure of proton-bound EmrE

EmrE monomers in the dimer are asymmetric relative to each other and display separate sets of signals in the NMR spectrum (Supplementary Fig. 1b). Exchange between these two conformations corresponds to the conformational switch between inward-open and outward-open states and impedes the collection of unambiguous distance constraints for structure determination using NMR spectroscopy[24,25,33]. To overcome this challenge for structural characterization, we developed a heterodimer approach involving a mixture of wild-type EmrE and the single-site mutant L51I (EmrE$^{L51I}$)[39]. Isotopically labeling one monomer in the dimer increased spectral resolution and allowed monomer-specific assignments by minimizing conformational exchange (Fig. 2a). Using EmrE-EmrE$^{L51I}$ heterodimer samples, we collected intramolecular and intermolecular distance constraints with solution NMR in DMPC/DHPC lipid bicelles and magic-angle-spinning in DMPC lipid bilayers. A total of 948 distance restraints were obtained from paramagnetic relaxation enhancement (PRE), nuclear Overhauser effect (NOE), and DARR/PDSD experiments (Supplementary Figs. 2–4 and Supplementary Table 1). In addition, we assigned 174 angular restraints from PISEMA experiments in aligned DMPC/DHPC lipid bicelles by acquiring $^{15}N$ chemical shifts and $^1H$-$^{15}N$ dipolar couplings (Supplementary Fig. 5). The NMR restraints were used to calculate an initial ensemble of structures, followed by refinement of the five lowest energy structures in a DMPC lipid bilayer in the absence of restraints. Several NMR-guided simulation rounds were performed by selecting structures from trajectories in best agreement with experimental data (Supplementary Fig. 6a). The final ensemble of 10 structures deposited in the Protein Data Bank reflects those from the MD simulation with the fewest violations of

experimental constraints (Supplementary Fig. 6b). Notably, more than 95% of the restraints were satisfied; additional details of the approach are provided in the "Structure determination" section of the "Methods".

The proton-bound structure of EmrE displays an anti-parallel arrangement of the dimer where the substrate binding pocket is formed by TM1, TM2, and TM3 (Fig. 2b). The tertiary fold positions TM1 between TM2 and TM3 on each monomer, resulting in the adjoining loop between TM2 and TM3 (loop 2) in proximity with N-terminal residues of TM1. The two Glu14 residues in TM1 each form intramolecular hydrogen bonds with the backbone carbonyl of Trp63 in TM3 and intermolecular hydrogen bonds with Tyr60 in TM3 of the opposite monomer (Fig. 2c). Accompanying these contacts are indole intermolecular interactions between the hydrophobic face of Trp63 that create a gate-like arrangement (Fig. 2c, d). Hence, interactions involving Glu14, Tyr60, and Trp63, highly conserved residues in the SMR family[40], as well as a hydrogen bond between Ser43 and Trp63 in monomer A, effectively occlude the substrate binding pocket. The buried locations of Glu14 in the hydrophobic pocket and its hydrogen bonding interactions with the backbone of Trp63 are likely responsible for the elevated p$K_a$ values of 7.2 and 8.4 for monomer A and monomer B, respectively[36]. The distance between Glu14 residues ranges from 8.6 Å to 9.4 Å among the structural ensemble and is consistent with the observed independent p$K_a$ values indicating a lack of observed electrostatic coupling[36]. Solvent accessibility to the Glu14 residues is expected to be somewhat limited, although there are small entrances available to the pocket via the membrane normal direction and between the two TM2 helices (Fig. 2b). Lastly, it is notable that the location of the Ile51 mutant in monomer A of the heterodimer faces toward the substrate binding pocket and is proximal to a hydrophobic latch comprised of Leu7 in TM1 and Ile54 in loop 2. Due to the asymmetry of contacts within the dimer it is likely the mutant has more favorable packing in this side of the dimer relative to the native Leu51 residue in monomer B.

Consistent with our Trp63 measurements (Fig. 1 and Supplementary Fig. 1c), the NMR structure of proton-bound EmrE revealed two key differences to a structure of EmrE crystallized at pH 5.2 and purported to be in the proton-bound conformation. First, the crystal structure displayed the side chains of Glu14 oriented toward an embedded water molecule in the substrate binding pocket, which differed from its proximity to the Trp63 backbone carbonyls in our NMR structure (Supplementary Fig. 7a). Second, the X-ray structure displayed each Trp63 indole amine oriented toward the substrate binding pocket of the transporter and in proximity to hydrogen bond with the embedded water molecule. Namely, the chi2 rotamers of the Trp63 residues deviate from our NMR structure by ~180° (Supplementary Fig. 7b). To better understand the source of these differences, we performed MD simulations on our NMR-derived structure and the X-ray structure where both Glu14 residues were protonated. Simulations of our NMR-derived proton-bound structure displayed stable conformations of Glu14 and Trp63 for each replicate simulation, while simulations of the X-ray structure displayed a flipped Trp63 conformation of monomer A for one of the three replicate simulations that resulted in a similar conformation as our NMR structure (Supplementary Fig. 7c–f, left). Following the Trp63 flip, the conformer remained stable for the remainder of the simulation (-1.9 μs). These simulation results support the conclusion that our NMR-derived structure represents the lowest energy conformation of proton-bound EmrE.

## TPP binding position clashes with the proton-bound structure

To gain insight into a drug bound conformation for comparison with our proton-bound structure of EmrE, we solved a structure of TPP-bound EmrE under similar experimental conditions as the proton-bound structure (Supplementary Fig. 6). The difference in the

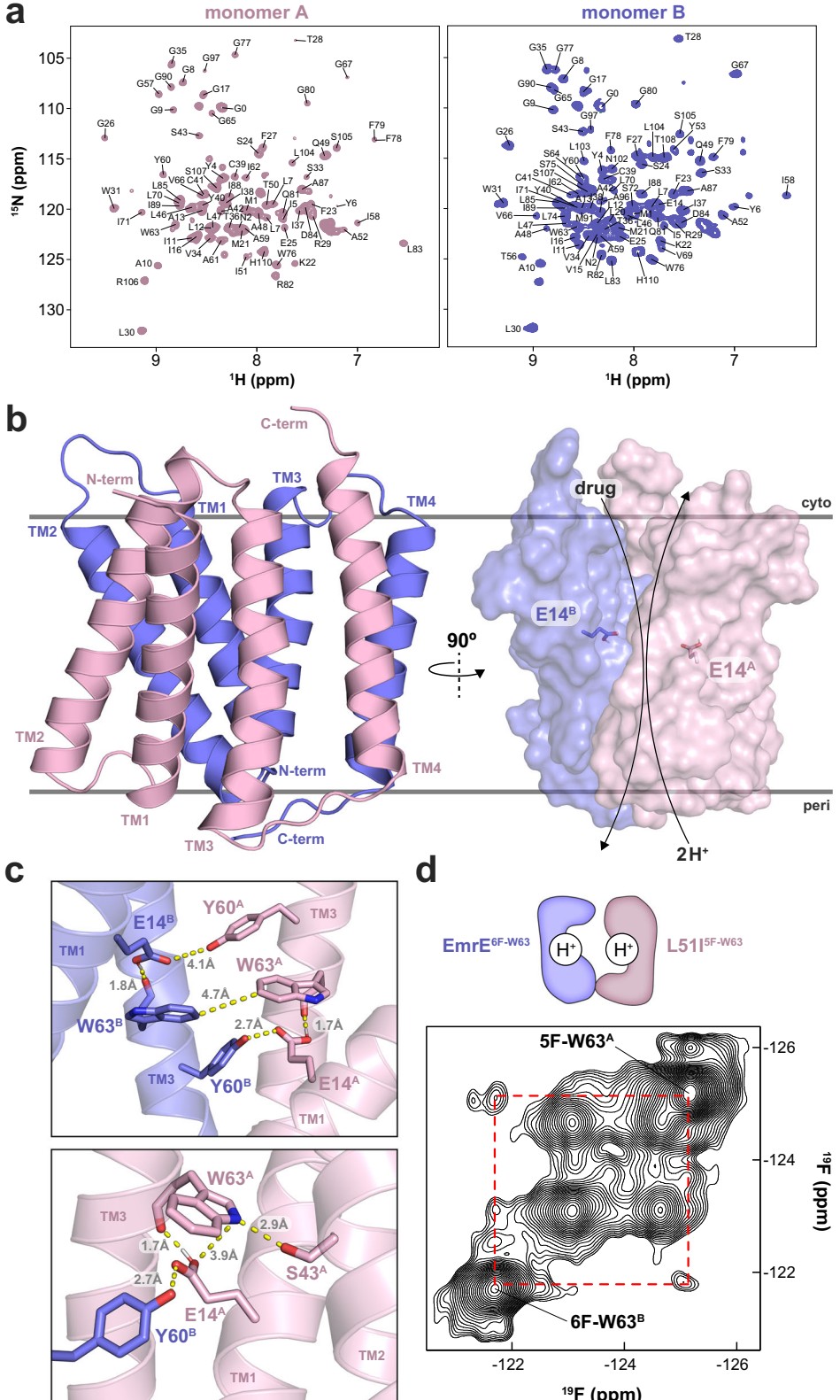

**Fig. 2 | Proton-bound EmrE structure determined using NMR spectroscopy.**
**a** ¹H/¹⁵N TROSY spectra of EmrE$^{L51I}$-EmrE heterodimers used for structural studies.
Left: ¹⁵N-labeled EmrE$^{L51I}$ mixed with natural abundance EmrE, corresponding to
monomer A peaks in pink. Right: ¹⁵N-labeled EmrE mixed with natural abundance
EmrE$^{L51I}$, corresponding to monomer B peaks in blue. **b** NMR structural repre-
sentations of the EmrE$^{L51I}$-EmrE heterodimer from the side view of the membrane
displaying monomer A in pink and monomer B in blue. The partially transparent
surface representation on the right displays Glu14 residues as sticks from monomers
A and B. In both views, the closed side of the transporter is displayed on the

periplasmic side of the membrane. **c** Zoom in views of the EmrE$^{L51I}$-EmrE hetero-
dimer displaying monomer A in pink and monomer B in blue. Dashed yellow lines
correspond to distances (in Å). Superscripts "A" or "B" refer to the corresponding
monomer A or B, respectively. **d** ¹⁹F/¹⁹F NOESY NMR spectra at a mixing time of 500
msec where the heterodimer sample was comprised of 5-fluoro-Trp63 labeled
EmrE$^{L51I}$ mixed with 6-fluoro-Trp63 labeled EmrE. Each EmrE monomer contained
only a single tryptophan residue (i.e., W31F/W45F/W76F). The dotted red box indi-
cates the intermolecular NOE. "H⁺" represents protonated Glu14 residues in the
cartoon schematic.

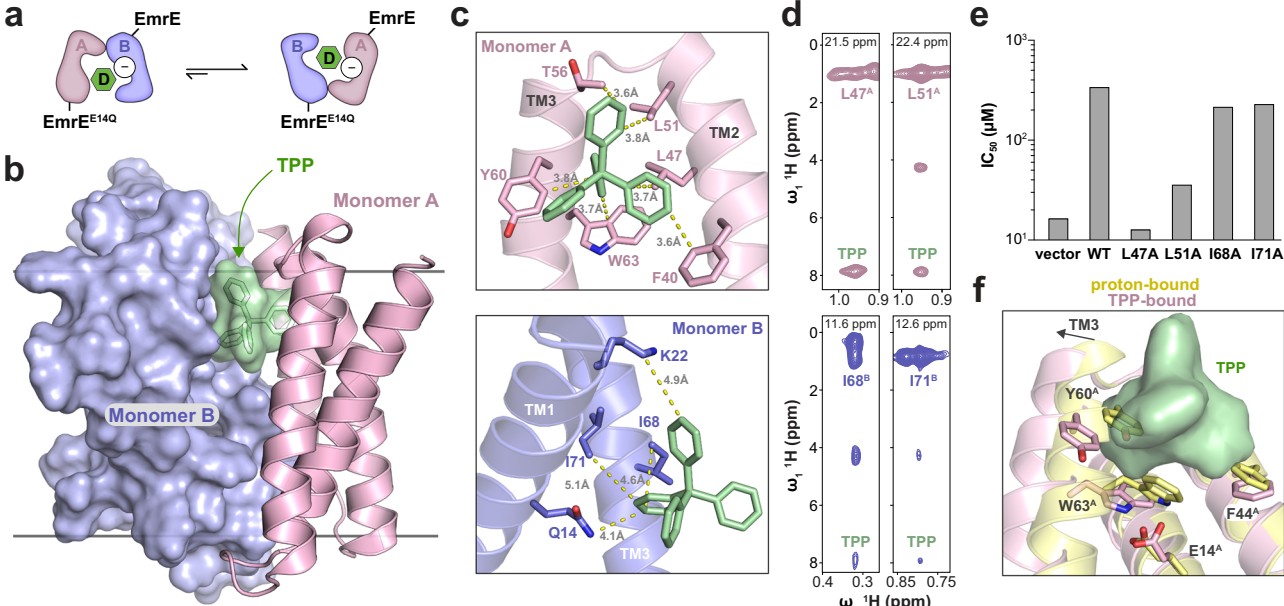

**Fig. 3 | Structure of TPP-bound EmrE determined using NMR spectroscopy.**
**a** Schematic displaying how the EmrE-EmrE$^{E14Q}$ heterodimer has a biased equilibrium where EmrE favors monomer A and EmrE$^{E14Q}$ favors monomer B. "D" denotes TPP (in green) and "–" indicates deprotonated Glu14 residues. **b** NMR structure of a side view of EmrE-EmrE$^{E14Q}$ heterodimer displaying monomer A in a cartoon representation (pink), monomer B in a surface representation (blue), and TPP in a stick and transparent surface representation (green). **c** Zoom in views of the EmrE-EmrE$^{E14Q}$ heterodimer making asymmetric contacts with TPP (green). Monomers A (pink, top) and B (blue, bottom) are displayed in a cartoon representation with side chains in sticks. Dashed yellow lines correspond to distances (in Å) between residues on EmrE and TPP. **d** NOE strip plots from a three-dimensional $^1$H/$^1$H/$^{13}$C HSQC-NOESY spectrum acquired on EmrE-EmrE$^{E14Q}$ heterodimers bound to TPP at pH 5.8 where either EmrE (pink spectra, top) or EmrE$^{E14Q}$ (blue spectra, bottom) was isotopically enriched with $-^{13}$CH$_3$ methyl groups at isoleucine, leucine, and valine.

Strips correspond to $^1$H/$^1$H spectral slices at the indicated $^{13}$C frequency in the indirect dimension. Labeled "TPP" cross-peaks indicate intermolecular NOEs between monomer A or B and TPP (superscripts A or B refer to the corresponding monomer). **e** Growth inhibition experiments against ethidium bromide in liquid culture of *E. coli* expressing wild-type EmrE ("WT") or the indicated mutants. An empty vector was used as the control ("vector"). The plotted $IC_{50}$ value was obtained by fitting the OD$_{600nm}$ at the 24 h timepoint as a function of variable ethidium bromide concentrations. Data are presented as mean values of the fitted $IC_{50}$ value from four replicates encompassing at least two independent experiments. **f** Superimposition of proton-bound and TPP-bound EmrE heterodimer structures displayed in yellow and pink cartoon representations, respectively. The view highlights the positions of select residues in monomer A relative to TPP (in green).

structure determination approach from proton-bound EmrE was to leverage a heterodimer of wild-type EmrE and the E14Q mutation (EmrE$^{E14Q}$) for some of the experimental restraints (Supplementary Figs. 3–5 and 8). This heterodimer mimicked the drug-bound conformation where Glu14 of monomer A was deprotonated and Glu14 of monomer B was protonated, which represents a pertinent state in the catalytic cycle of EmrE (Fig. 3a and Supplementary Fig. 6a)[30,36]. Within these samples, monomer A or B was isotopically enriched in the EmrE-EmrE$^{E14Q}$ heterodimer, leading to simplified NMR spectral analyses[36].

The TPP-bound EmrE structure displays an open conformation where the substrate binding pocket is accessible to drug from one side of the membrane and forms asymmetric contacts with residues in monomers A and B (Fig. 3b, c). Most of the closest interactions are made with monomer A and include aromatic residues Phe44 of TM2 and Tyr60 and Trp63 of TM3. Mutation of these sites diminishes drug binding and displays loss-of-function in resistance assays[41,42]. Additional TPP interactions were made with hydrophobic residues in EmrE, which were supported by NOEs to TPP, including Leu47 and Leu51 in TM2 of monomer A and Ile68 and Ile71 in TM3 of monomer B (Fig. 3d). Mutation of these residues displayed loss-of-function, in agreement with their position in the substrate binding pocket (Fig. 3e)[43–45]. Namely, L47A and L51A mutations showed ablated or near ablated activity, while I68A and I71A mutations displayed a ~2-fold reduced $IC_{50}$ relative to wild-type EmrE. The latter observations were consistent with rather weak NOEs and a corresponding greater distance to TPP relative to Leu47 and Leu51 of monomer A (Fig. 3d).

Three key observations were noted when comparing our TPP-bound and proton-bound EmrE structures. First, the location of TPP in the pocket clashed with the positions of Phe44, Tyr60, and Trp63 of monomer A in the proton-bound structure (Fig. 3f). This observation was in agreement with the occluded nature of the substrate binding pocket for proton-bound EmrE. Second, superimposition of the structures displayed a ~3 Å movement of the N-terminal portion of TM3 away from the substrate binding pocket for the TPP-bound structure. Underlying this conformational change was the loss of hydrogen bonds between the carboxyl of Glu14 and the backbone of Trp63 in the TPP-bound structure. This observation underscored our finding that Glu14 interactions latch TM3 in monomer A to occlude the binding pocket in proton-bound EmrE. Notably, hydrogen bonds involving Glu14 to the backbone of TM3 were also absent in monomer B of the TPP-bound structure. This change likely occurred due to the interaction observed between Gln14 (mutant) with TPP in the substrate binding pocket which may explain the more acidic p$K_a$ value for Glu14 in monomer B upon TPP binding[30]. Third, the indole orientation of Trp63 in TM3 of monomer A had a change in the side chain rotamer relative to proton-bound EmrE (Fig. 3f). This structural change was consistent with the 100 ppm $^{15}$N chemical shift change of the indole nitrogen of Trp63 in oriented sample solid-state NMR experiments (Fig. 1b, c). Hence, it appears that the conformational movement of TM3 within monomer A, including that of the Trp63 side chain, serves as a *gatekeeper* for accessing the drug bound conformation.

## The protonation state of Glu14 in monomer B modulates the TPP binding location

Since EmrE binds TPP in the fully apo conformation and when only one of the two Glu14 residues is protonated[30], we compared our TPP-bound structure with those previously reported in complex with TPP. We found our structure displays 3.4 Å and 3.6 Å backbone r.m.s.d. relative to NMR structures bound to fluorinated TPP at pH 5.8 (PDB ID 7JK8)[29] and pH 8.0 (PDB ID 7SFQ)[28], and 2.0 Å backbone r.m.s.d. relative to an X-ray structure bound to TPP at pH 7.25 (PDB ID 7SV9)[31] (Supplementary Fig. 9a). The TPP location in the substrate binding pocket differed on average by ~2.5 Å from the prior NMR structure at pH 5.8, ~4.2 Å from the prior NMR structure at pH 8.0, and ~7.6 Å from the X-ray structure. We also observed rotamer changes for Trp63 of monomer B with respect to each structure. The relatively high backbone r.m.s.d and deviations in the TPP location from the prior NMR structural work may arise from the reliance on a Cα model crystal structure of EmrE (PDB ID 3B5D) to initiate the structure determination process and potential electrostatic differences between protonated and fluorinated TPP. Differences with the TPP binding position relative to the X-ray structure (PDB ID 7SV9), including a rotamer change for Trp63 of monomer B, may reflect a difference in Glu14 protonation within monomer B.

To gain experimental insight into the role of Glu14 protonation of monomer B, we collected NMR spectra over a range of pH values for wild-type EmrE bound to TPP (Supplementary Fig. 9b–d). These conditions enabled acid/base chemistry on Glu14 of monomer B while maintaining TPP binding and Glu14 of monomer A in the deprotonated state[30]. Solution NMR spectra revealed a large spectral perturbation for Ile68 of monomer B, which was consistent with an expected change in the electrostatic environment around this residue upon movement of TPP (Supplementary Fig. 9d). Namely, Ile68 is ~4 Å from TPP in our NMR structure and ~8 Å from TPP in the X-ray structure (Supplementary Fig. 9e). In addition, MD simulations performed on our TPP-bound structure of EmrE revealed TPP experienced relatively small movements from its starting position and no deeper insertion into the binding pocket as observed in the X-ray structure (Supplementary Fig. 9f). Based on these findings, we conclude that protonation of Glu14 of monomer B modulates the TPP binding location, similar to a previous report[28], and that the X-ray TPP-bound EmrE structure[31] is likely deprotonated at both Glu14 residues.

## MD simulations support a conformational selection model

NMR spectroscopic observations suggested deprotonation of Glu14 in monomer A induced a conformational change leading to drug accessibility of the substrate binding pocket. To test this hypothesis, we performed MD simulations on our NMR structures (proton-bound and TPP-bound states) and ones where Glu14 of monomer A was deprotonated (starting from the proton-bound NMR structure). As noted above, simulations on proton-bound EmrE displayed no significant deviations from those observed within the NMR-derived ensemble, including stable hydrogen bonds between Glu14 carboxyl sites and the Trp63 backbone carbonyls (Fig. 4a). Furthermore, no water molecules were found within 5 Å of Glu14 of monomer A, thereby preserving the occluded conformation of Glu14 in monomer A from solvent (Fig. 4b, c). In contrast, simulations of deprotonated Glu14 in monomer A displayed an altered position of the glutamate side chain, such that the carboxylate group oriented toward the substrate binding pocket and away from its position in a hydrogen bond with the backbone carbonyl of Trp63 (Fig. 4d and Supplementary Fig. 7c, e, right). Correspondingly, we observed an increase of water entering the pocket, including solvation of Glu14 of monomer A in its altered conformation (Fig. 4e, f)[46]. Simulations starting from our TPP-bound structure revealed that Glu14 of monomer A remained in an orientation toward the substrate binding pocket and in a similar conformation as

deprotonated EmrE (Fig. 4g). Several water molecules resided within 5 Å of Glu14 in monomer A in this drug-bound conformation (Fig. 4h, i).

For Trp63, simulations displayed stable rotamers for the proton-bound state (Fig. 4a), which was consistent with the homogeneous Trp63 signals observed in MAS correlation spectra and the proximity between the Trp63 residues on the hydrophobic face of the indole ring (Figs. 1 left and 2c). However, simulations where Glu14 of monomer A was deprotonated showed that the side chain Trp63 rotamer of monomer A occupied two chi2 rotamers (Fig. 4d). Notably, the same occurrence was seen in simulations of the EmrE X-ray structure (PDB ID 7SV9) where Glu14 was deprotonated in monomer A (Supplementary Fig. 7d, f, right). Of the two chi2 rotamers detected in monomer A, one resembled that of the proton-bound conformation while the other was flipped by ~180° and resembled the TPP-bound conformation (Fig. 4g). These simulation results on deprotonated EmrE correlated with the heterogeneity observed in MAS experiments for the side chain of Trp63 (Fig. 1a, middle). The flipped conformation of Trp63 of monomer A oriented the amine group of the indole toward the substrate binding pocket, resulting in hydrogen bond formation with water. This result provided an explanation for the ~10 ppm $^{15}N$ chemical shift change observed in MAS experiments (Fig. 1a, middle), which has previously been correlated to hydrogen bond formation in tryptophan indole side chains buried in the hydrophobic core of soluble proteins[47]. The conformational change of Trp63 is also consistent with the reduced quantum yield of the Trp63 fluorescence spectrum for the proton-bound state relative to the deprotonated state (Supplementary Fig. 1c) and is reminiscent of the orientations of Glu14 and Trp63 in the drug-free X-ray structure (PDB ID: 7MH6). Based on these findings, it is likely the drug-free X-ray structure corresponds to Glu14 in a deprotonated state. Overall, these simulation results support a conformational selection drug binding mechanism that is modulated by deprotonation of Glu14 in monomer A.

## Discussion

Four of the five multidrug efflux families carry out secondary active transport by harnessing the PMF or differences in solute concentrations across the membrane. Antiporters like EmrE rely on the alternating access mechanism to transport substrates and protons by switching between inward-facing and outward-facing directions. In fact, 12-TM domain transporters from the Major Facilitator Superfamily (MFS) and others occupy additional conformations along the transport cycle, including inward-open, inward-occluded, occluded, outward-occluded, and outward-open states[16,48,49]. Switch-like mechanisms have been proposed for MFS family members where protonation of membrane embedded aspartate and glutamate residues regulate opening and closing of the substrate binding pocket. Recent findings on QacA[50], MdfA[51], and NorA[52] describe how protonation modulates conformational changes between such inward- and outward-facing states. Measurements sensitive to dynamics, such as single molecule fluorescence experiments, NMR spectroscopy, and EPR spectroscopy, have complemented structural findings by showing how substrate and ion binding modulate the rate of conformational switching between inward- and outward-facing conformations[33,53,54].

The novelty of this work is the simultaneous characterization of structure and conformational heterogeneity under well-defined Glu14 protonation states of EmrE. Our findings reveal how deprotonation of the membrane embedded Glu14 residue in monomer A disrupts a hydrogen bond between its carboxyl group and the backbone carbonyl of Trp63 in TM3 (Fig. 5). Disruption of this interaction induces the side chain of Trp63 in monomer A to populate two states that resemble the occluded proton-bound conformation and the open TPP-bound conformation. Substrates bind to the latter conformation, indicating a molecular recognition mechanism involving conformational selection. Hence, acid/base chemistry at Glu14 of monomer A modulates an equilibrium change toward the open state of the transporter. This

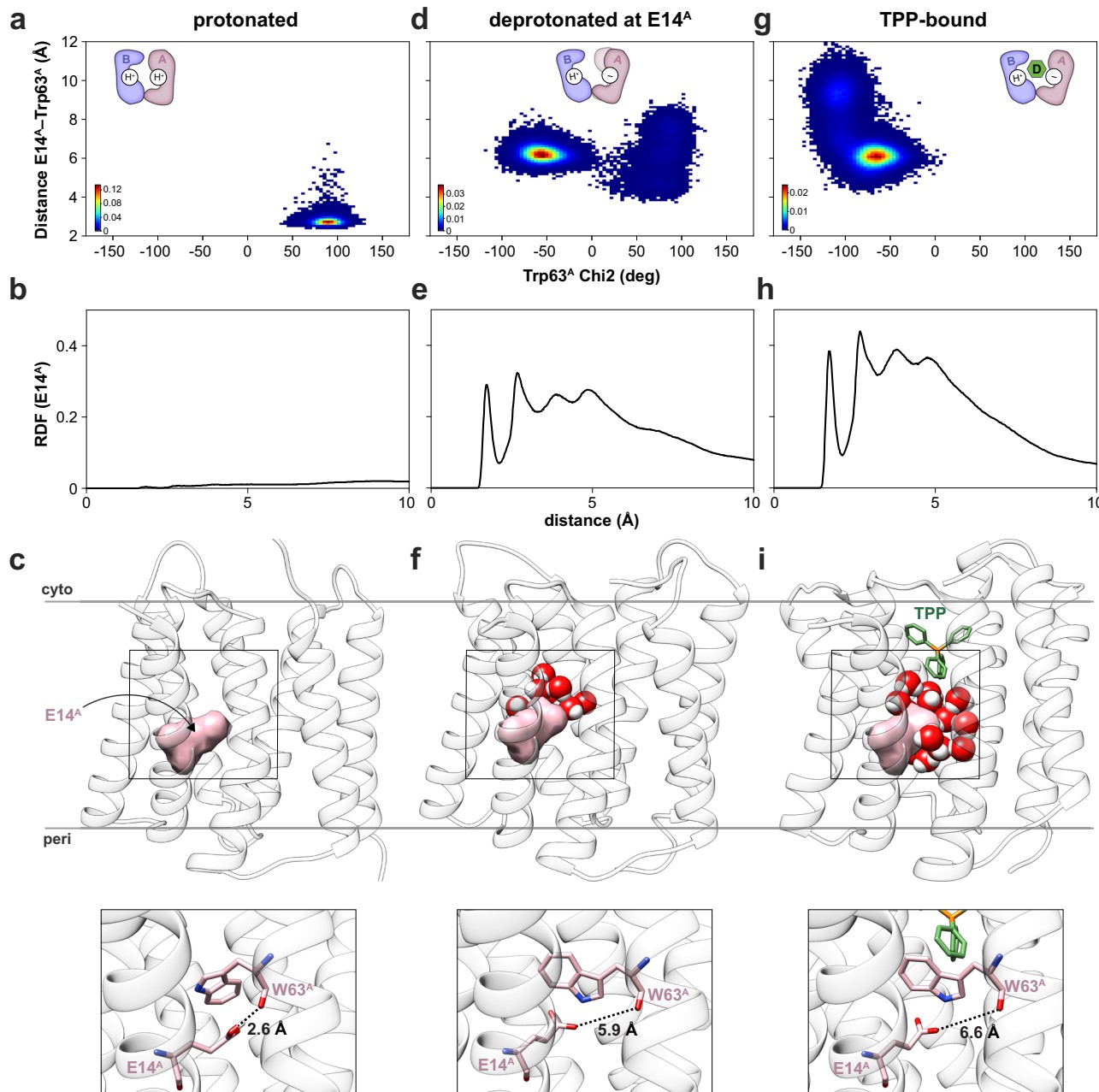

**Fig. 4 | Protonation of Glu14 of monomer A induces the conformational selection mechanism.** Heat map plots displaying the distance between the side chain carboxyl oxygen of Glu14[A] and the backbone carbonyl oxygen of Trp63[A] against the chi2 angle of Trp63[A] (superscripts refer to monomer A) obtained from MD simulations for proton-bound EmrE (**a**), EmrE deprotonated at Glu14 of monomer A (**d**), and TPP-bound EmrE (**g**). Monomer A is represented in pink in the dimeric cartoon, while monomer B is colored in blue. "H+" and "−" denote protonated or deprotonated Glu14 residues. Water radial distribution function (RDF) surrounding Glu14 of monomer A derived from MD simulations on proton-bound EmrE (**b**), EmrE deprotonated at Glu14 of monomer A (**e**), and TPP-bound EmrE (**h**).

Representative snapshots derived from MD simulations on proton-bound EmrE (**c**), EmrE deprotonated at Glu14 of monomer A (**f**), and TPP-bound EmrE (**i**). Water molecules within 5 Å from Glu14 of monomer A are represented by red (oxygens) and white (hydrogens) spheres. Glu14 of monomer A is highlighted by pink surfaces, and TM helices are colored in white. The insets on the bottom display expanded views of the boxed portions from the top views. Glu14 and Trp63 of monomer A are displayed in sticks and dashed black lines display the distance between the carboxyl of Glu14 and the backbone of Trp63 (in Å). Waters are not displayed in the expanded view.

model provides an explanation to the pH dependence of drug binding (Supplementary Fig. 1a). Such an experimental structure of an occluded conformation we determined for the proton-bound state has not previously been reported for EmrE or another SMR family transporter. However, the presence of an occluded conformation for EmrE was proposed based on distance measurements from EPR spectroscopy[55] and MD simulations[46]. We hypothesize our structure is significant since it likely serves as an intermediate conformation between outward-

open and inward-open conformations, resulting in the movement of protons from the periplasm to the cytoplasm. This interpretation is consistent with faster alternating access exchange for proton-bound EmrE compared to deprotonated states of EmrE at higher pH values[33].

What additional structural evidence supports the conformational selection model? Guided by NMR measurements directly probing Trp63 within the substrate binding pocket, we propose that the drug-free X-ray structure of EmrE[31] likely corresponds to one of the

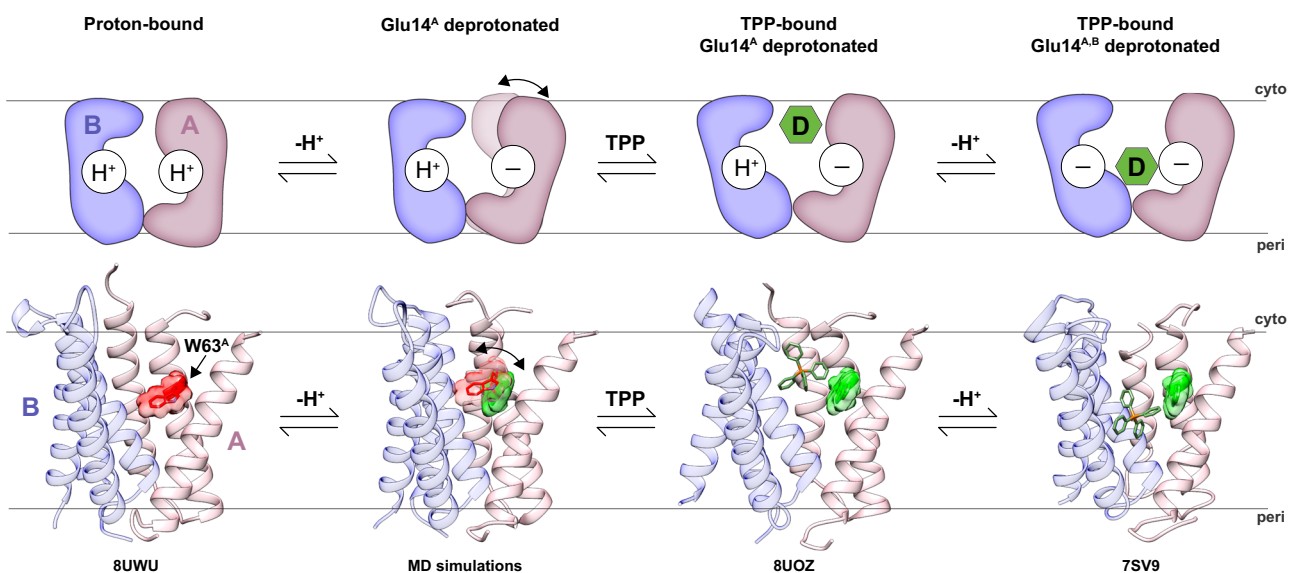

**Fig. 5 | EmrE molecular recognition model.** Cartoon (top row) and structural models (bottom row) for how EmrE binds TPP on the cytoplasmic side of the membrane. Monomers A and B are colored in pink and blue, "H+" and "−" denote protonated or deprotonated Glu14 residues, "D" corresponds to TPP (in green), and Trp63 of monomer A is displayed in sticks and surface representations in the bottom row (red is occluded conformation; green is open conformation). In this model, deprotonation of Glu14 of monomer A induces a change in Trp63 of monomer A, resulting in conformational heterogeneity between occluded and open conformations (indicated by the double arrow). Subsequently, TPP binds to the open "green" conformation of Trp63 through conformational selection. TPP binding induces deprotonation of Glu14 in monomer B which favors TPP movement to the deeper binding position, a conformation likely poised for conformational exchange. Below the structural models are PDB IDs or an indication whether the model was derived from MD simulations. Note that TM2 of monomer A was removed for clarity.

conformations EmrE samples when Glu14 of monomer A becomes deprotonated. Indeed, the presence of a water molecule in the substrate binding pocket of this crystal structure is consistent with MD simulations showing that deprotonation of Glu14 in monomer A leads to greater water penetration into the pocket. Hence, analyses of NMR spectroscopy and MD simulations in this work together with X-ray crystallographic findings[31] enabled a more complete structural basis of drug binding (Fig. 5).

In conclusion, our results provide a model for how dynamics are modulated upon deprotonation of a single glutamate residue within the EmrE dimer. The corollary is that acid/base chemistry of membrane embedded acidic residues serves as a trigger for equilibrium changes, not conformational switching reported in other efflux systems. We propose that substrates bind to the open-like conformation of the equilibrium through conformational selection, ultimately resulting in a shift toward the substrate-bound state through Le Chatelier's principle. The presence of glutamate and aspartate residues commonly found in other proton-coupled transporters suggests ionization of membrane embedded acidic residues could be a common mechanism for modulating equilibrium shifts and the accessibility of the substrate binding pocket.

## Methods

### EmrE expression and purification

EmrE is fused to the C-terminal end of His-tagged maltose binding protein (MBP) and encoded in a pMal c2x vector (New England Biolabs). After transformation in BL21(DE3) *E. coli*, preparation of protein at natural abundance involves growing bacteria in lysogeny broth (LB) at 37 °C. Expression of the fusion protein is induced by addition of 1 mM IPTG to the media at an $OD_{600nm}$ value of ~0.8 to 1.0. After IPTG induction, the culture is grown for 16–20 h at a temperature of 20 °C. Next, cultures are harvested by centrifugation and stored at −80 °C. Typically, a 2.5 l growth of bacteria is lysed in 600 ml of the following buffer: 20 mM phosphate buffer pH 7.3, 120 mM NaCl, 1 mM EDTA, 0.1 mM DTT, 0.5% glycerol (v/v), 0.5 μg/ml pepstatin A (w/v), 0.5 μg/ml

leupeptin (w/v), 80 μg/ml lysozyme (w/v), 1% Triton X-100 (v/v), and 0.5 mM PMSF. The lysate is sonicated on ice for 30 min and subsequently centrifuged for 20 min at $48,384 \times g$ using a Beckman centrifuge equipped with a JA-25.50 rotor at 4 °C (Beckman Coulter). Next, the supernatant is loaded on an amylose resin (New England Biolabs) and washed with the following buffer: 20 mM phosphate buffer pH 7.3, 120 mM NaCl, 0.02% n-dodecyl-β-D-maltopyranoside (DDM, Anatrace), and 250 μM DTT. The fusion protein is eluted with 20 mM phosphate buffer pH 7.3, 120 mM NaCl, 0.02% DDM, 250 μM DTT, and 20.85 g/l maltose (w/v). The eluate is concentrated to ~1 mg/ml and cleaved using TEV. Note that the TEV cleavage site, located between MBP and EmrE, results in three non-native Gly residues on the N-terminal side of EmrE. Following cleavage, the solution is passed over a Ni-NTA resin (Thermo Fisher Scientific) to bind His-tagged MBP and TEV. The flow through, that predominantly contains EmrE, is concentrated and purified using size exclusion chromatography in 20 mM phosphate buffer pH 7.3, 120 mM NaCl, and 0.08% DDM.

For purifications involving isotopically labeled proteins, an identical protocol was used. However, the growth media is varied to incorporate ${}^{13}C$, ${}^{15}N$, ${}^{2}H$, and ${}^{19}F$, and is described as follows. For solution NMR experiments, ${}^{2}H/{}^{15}N/{}^{13}C$ proteins were expressed in M9 media in $D_2O$ with addition of 0.2% ${}^{2}H/{}^{13}C_6$ D-glucose (w/v), 2 mM MgSO4, 0.1 mM $CaCl_2$, vitamins, and minerals. The M9 media is comprised of 3 g/l $KH_2PO_4$ (w/v), 12.8 g/l $Na_2HPO_4$ (w/v), 0.5 g/l NaCl (w/v), and 1 g/l ${}^{15}N$ $NH_4Cl$ (w/v). Preparation of selectively labeled Ile, Leu, and Val methyl samples, such as [Ile-${}^{13}C\delta H_3$, U-${}^{15}N$, ${}^{2}H$] and [Leu,Val-${}^{13}C_\delta H_3$ or ${}^{13}C\gamma H_3$, U-${}^{15}N$, ${}^{2}H$]-labeled EmrE, were expressed in the above M9 growth media (${}^{2}H$ D-glucose instead of ${}^{2}H/{}^{13}C$ D-glucose) by addition of 50 mg/l 2-ketobutyric acid-4-${}^{13}C$, 3,3-${}^{2}H_2$ sodium salt hydrate and 80 mg/l alpha-ketoisovaleric acid -3-methyl-${}^{13}C$, 3,4,4,4-${}^{2}H_4$, respectively. ${}^{19}F$ proteins were prepared by addition of all amino acids (300 mg/l of unlabeled amino acids, 800 mg/l of amino acids that the labeled one scrambles to, and 60 mg/l ${}^{19}F$ labeled fluoro indole) in M9 media. 5 or 6-fluoroindole was added 1 h prior to induction. Bacteria were allowed to grow only ~6 h after induction at 20 °C to prevent scrambling.

 

For oriented sample solid-state NMR, $^{15}$N labeled selectively labeled samples ($^{15}$N-Ile, $^{15}$N-Val, $^{15}$N-Leu, $^{15}$N-Tyr, $^{15}$N-Met, $^{15}$N-Thr, $^{15}$N-indole, and $^{15}$N-Trp)[56] were prepared by addition of all amino acids (300 mg/l of unlabeled amino acids, 800 mg/l of amino acids that the labeled one scrambles to, and 120 mg/l $^{15}$N labeled amino acid) in M9 media without ammonium chloride (3 g/l KH$_2$PO$_4$ (w/v), 12.8 g/l Na$_2$HPO$_4$ (w/v), 0.5 g/l NaCl (w/v). The $^{15}$N labeled amino acid was added ~45 min prior to induction. Bacteria were grown for ~4 h after induction at 25 °C to prevent scrambling. The indole labeling was accomplished by adding in $^{15}$N indole (60 mg/l) and natural abundance serine (60 mg/l) to prevent scrambling.

For MAS experiments for distance restraints, EmrE was grown in 2-$^{13}$C or 1,3-$^{13}$C glycerol labeling. For 2-$^{13}$C glycerol labeling, EmrE was grown using 2-$^{13}$C glycerol (4 g/l) in the following media for 12 h: 3 g/l KH$_2$PO$_4$, 12.8 g/l Na$_2$HPO$_4$, 0.5 g/l NaCl, 1.0 g/l $^{15}$NH$_4$Cl, and 2 g/l NaH$^{13}$CO$_3$. For 1,3-$^{13}$C glycerol labeling, EmrE was grown using 1,3-$^{13}$C glycerol (4 g/l) in the following media: 3 g/l KH$_2$PO$_4$, 12.8 g/l Na$_2$HPO$_4$, 0.5 g/l NaCl, 1.0 g/l $^{15}$NH$_4$Cl, and 2 g/l NaHCO$_3$ at natural abundance. For dynamic nuclear polarization (DNP) experiments to obtain constraints to $^{13}$C-labeled TPP, we prepared $^{13}$C$^{\alpha,\beta}$-Tyr labeled EmrE. This labeling was prepared in the same manner as for selectively labeled oriented samples described above.

## Isothermal titration calorimetry

Purified EmrE samples were treated with 5 mM DTT and DDM was added to obtain a minimum molar ratio of 200:1 DDM:EmrE. The sample was dialyzed against 50 mM Na$_2$HPO$_4$ and 50 mM NaCl at the desired pH for 2 h in 10 kDa dialysis tubing. EmrE was concentrated using a 10 kDa cutoff centrifugal concentrating membrane in a Beckman rotor equipped with a JS 4.3 rotor spinning at 2799 × $g$. The resulting sample, ~450 µl, was injected into the sample cell of a low volume Nano ITC instrument from TA Instruments. TPP solutions were prepared by matching the DDM concentration of the same protein sample. All experiments were performed at 25 °C with 350 r.p.m. stirring. To obtain the heat of dilution, extra injections were carried out after the protein was saturated and these last injections were averaged to obtain the heat of dilution. Following each run, the sample was removed and the pH was checked to confirm minimal drift. In each case, the pH drift was within ±0.05 units. Binding affinities and thermodynamic parameters were determined using TA Instruments NanoAnalyze Data Analysis software (version 3.12.5) with an independent binding site model. Each binding experiment at different pH values was tested in two or more independent experiments. The reported mean is the average between these independent experiments.

## Solution NMR spectroscopy

For solution NMR studies, purified EmrE in DDM from SEC was reconstituted in dimyristoyl-sn-glycero-3-phosphocholine (DMPC) with the lipid chains perdeuterated (14:0 PC D$_{54}$, Avanti Polar Lipids) at a ratio of 1/1.6 (w/w) of protein to lipids. DDM detergent was removed from the sample by addition of Bio-Beads at a ratio of 90/1 (w/w) of the beads to detergent. Proteoliposomes were pelleted by ultracentrifugation at 347,500 × $g$ using a Beckman Optima MAX-XP benchtop ultracentrifuge equipped with a TLA-110 rotor (Beckman Coulter). The pellet was resuspended in a solution of dihexanoyl-sn-glycero-3-phosphocholine (DHPC) with the lipid chains perdeuterated (6:0 PC D$_{22}$) to form isotropic bicelles ($q$ = 0.3). The final NMR samples for solution NMR contained 0.5 mM EmrE, 150 mM Na$_2$HPO$_4$ pH 6.0, 20 mM NaCl, and 50 mM DTT. Heterodimer samples were prepared by mixing the two proteins, EmrE with EmrE$^{E14Q}$ or EmrE$^{L51I}$, at a molar ratio 1/1.6 or 1/1.8, where the protein in excess corresponded to the one without $^{15}$N or $^{13}$C labels (i.e., NMR silent). The two proteins used to prepare the heterodimer sample were incubated at 37 °C with stirring in the presence of 50 mM DTT for 1 h immediately prior to

reconstitution into lipids. Reconstitution into bicelles was carried out in the same manner as those for homodimer samples. For experiments on the TPP-bound form of EmrE, TPP was added at 8-fold greater concentration relative to the EmrE monomer.

Samples for paramagnetic relaxation enhancement (PRE) experiments were prepared similar to the protocol above. Single cysteine mutants of EmrE were introduced at positions 5, 30, 39, 41, or 95 in the primary sequence. In each construct, all native cysteines located at positions 39, 41, and 95 not subjected for labeling were mutated to serine. The purified protein was reduced with 10 mM DTT prior to the ((1-Oxyl-2,2,5,5-tetramethyl-Δ3-pyrroline-3-methyl) methanethiosulfonate) (MTSL) reaction. EmrE was buffer exchanged into 20 mM Na$_2$HPO$_4$ pH 7.4 and 40 mM NaCl to remove DTT. The MTSL tag was added in two equal increments to total ~120-fold excess concentration relative to the protein concentration. The labeling was monitored by MALDI-TOF to ensure efficient labeling (>90%). Once labeled the protein was buffer exchanged to remove any excess MTSL and heterodimer mixing was done without DTT at 37 °C. The reconstitution method was performed as described above.

Solution NMR experiments were performed with AVANCE III or NEO Bruker spectrometers operating at a $^1$H frequency of 600, 700, or 800 MHz each equipped with a triple resonance TCI cryogenic probe. All experiments were acquired at 310 K. Backbone assignment experiments involved a series of triple resonance experiments, including HNCA, HNCO, and HN(CO)CA. For backbone and methyl PRE measurements, $^1$H/$^{15}$N TROSY experiment and $^1$H/$^{13}$C HSQC experiment were run with a 4 s delay between experiments to measure intensity retention corresponding to residues affected by the paramagnetic tag (MTSL). Spectral widths were typically set 12,019.2 Hz and 1399.0 Hz for $^1$H and $^{15}$N in $^1$H/$^{15}$N TROSY experiments and 10,000.0 Hz and 2565.4 Hz for $^1$H and $^{13}$C in $^1$H/$^{13}$C HSQC experiments. Methyl residues in proximity with TPP were determined using three-dimensional NOESY-HSQC experiments with otherwise perdeuterated proteins and deuterated solvent to ensure NOE contacts observed were from TPP. Control experiments with fully deuterated TPP were performed and no NOE contacts were observed. Three-dimensional HMQC-NOESY-HMQC experiments with mixing times of 300 ms, 500 ms, and 1 s were performed on heterodimer samples of (EmrE-EmrE$^{E14Q}$ and EmrE$^{L51I}$-EmrE) to measure intramolecular and intermolecular contacts.

## Tryptophan fluorescence spectroscopy

Tryptophan fluorescence experiments were performed using an EmrE construct containing only one tryptophan at residue 63 (EmrE$^{W63}$; it contained mutations W31F/W45F/W76F). Isotropic bicelle samples were prepared similar to solution NMR samples where the molar ratio of DMPC and DHPC was 1/3. Freshly purified EmrE$^{W63}$ (1.8 mg) was reconstituted in DMPC (14.6 mg) at a protein to lipid ratio of 1/150 (mol/mol). Proteoliposome samples were split into two samples containing 100 mM Na$_2$HPO$_4$ and 100 mM NaCl buffers at either pH 5.0 or pH 9.0. Addition of DHPC was carried out to give a final volume of each sample of 1.5 ml. The final concentration of EmrE dimer in each sample was 24 µM. Fluorescence readings were performed on a Molecular Devices Flexstation 3 instrument using an excitation wavelength of 280 nm and an emission wavelength ranging between 250 nm and 400 nm. Each experiment was performed in triplicate and the error bars reflect the standard deviation between these replicates.

## Oriented sample solid-state NMR spectroscopy

For oriented sample solid-state NMR studies, SEC purified $^{15}$N labeled EmrE in DDM detergent was reconstituted into 1,2-di-O-tetradecyl-sn-glycero-3-phosphocholine/dihexanoyl-sn-glycero-3-phosphocholine (O-14:0-PC/6:0-PC) bicelles at a molar ratio of 3.5/1. The bicelles were made with a protein concentration ~2 mM and a total lipid concentration of 25% (w/v) in 80 mM HEPES and 20 mM NaCl at pH 6.0. A final concentration of 8 mM YbCl$_3$ was added to flip the orientation of

the bicelle such that the bicelle normal was parallel with the magnetic field. TPP was added at 8-fold the monomer concentration.

Experiments were acquired on an Agilent DD2 spectrometer operating at a $^1$H frequency of 600 MHz. PISEMA[57] spectra were acquired with SPINAL-64 decoupling[58] during acquisition and phase modulated Lee-Goldberg (PMLG)[59] in the indirect dimension. The effective radiofrequency field for SPINAL decoupling was 50 kHz, while the effective field for PMLG was 41 kHz. Spectra were acquired with ~1500 scans and 14 increments in the indirect dimension. The indirect dimension axes were corrected with the scaling factor of 0.82 and the $^{15}$N direct dimension was referenced to $^{15}$NH$_4$Cl (solid) at 41.5 ppm. A series of PISEMA experiments were collected on $^{15}$N amino acid labeled samples including Ile, Leu, Val, Tyr, Met, Thr, and Phe. The assignments of each amino acid were done by a combination of proton driven spin diffusion (PDSD) experiment, conformational biased heterodimer samples, and site directed mutagenesis.

## MAS solid-state NMR spectroscopy

For MAS solid-state NMR experiments, sequential assignments of monomer A and monomer B were performed using mixtures of EmrE$^{L51I}$ and EmrE for the proton-bound state or EmrE and EmrE$^{E14Q}$ for the TPP-bound state. In these samples, one protein was uniformly $^{13}$C/$^{15}$N labeled and the other was at natural abundance. Samples were prepared by mixing a ratio of 1/1.5 labeled/unlabeled EmrE in DDM at 37 °C prior to reconstitution. EmrE was reconstituted in O-14:0-PC at a protein to lipid ratio of 1/1 (w/w) with the total amount of protein present in each sample corresponding to ~10 mg. After removal of the detergent using Bio-Beads (Bio-Rad), proteoliposomes were centrifuged at 347,500 × $g$ using a Beckman Optima MAX-XP benchtop ultracentrifuge equipped with a TLA-110 rotor (Beckman Coulter). The pellet was buffer exchanged to 150 mM Na$_2$HPO$_4$ pH 5.0 and 20 mM NaCl and centrifuged again at 436,000 × $g$ using a Beckman Optima MAX-XP benchtop ultracentrifuge equipped with a TLA-100 rotor (Beckman Coulter). Each sample was transferred to a 3.2 mm Varian pencil-type rotor with spacers to prevent dehydration.

A series of triple resonance sequential experiments (NCACX, NCOCX, CAN(CO)CX, and CONCA) were acquired[60] at -273 K and a MAS rate of 12.5 kHz using an Agilent NMR spectrometer operating at a $^1$H frequency of 600 MHz equipped with a 3.2 mm triple resonance MAS probe (Black Fox, LLC). The typical 90° pulse lengths for $^1$H, $^{13}$C, and $^{15}$N nuclei were 2.5, 4.5, and 5 μs, respectively. For $^1$H-$^{13}$C/$^{15}$N cross-polarization, radiofrequency (RF) pulses of 55.6 kHz (or 50 kHz) were used for $^{13}$C (or $^{15}$N), respectively, with a tangent ramp applied on $^1$H[61]. Selective transfers were performed using SPECIFIC-CP[62] for transfers between $^{15}$N to/from $^{13}$CO and $^{15}$N to/from $^{13}$CA with a cross-polarization time of ~4–6 ms and with RF amplitudes of ~18.8 ($^{15}$N), ~31.3 ($^{13}$CA) or 48.9 ($^{13}$CO), and 110 kHz ($^1$H). Using sparsely labeled samples (2-$^{13}$C glycerol and 1,3-$^{13}$C glycerol), several two-dimensional PDSD or DARR[63] and three-dimensional NCACX experiments were acquired using mixing times ranging from 50 ms to 1 s. 100 kHz of $^1$H RF power was used for decoupling during both acquisition and evolution periods. Chemical shift referencing of $^{13}$C was performed by external referencing the CH2 resonance of adamantane to 40.48 ppm[64]; $^{15}$N referencing was done using the indirect reference from $^{13}$C.

Dynamic nuclear polarization (DNP) was performed to measure EmrE constraints to $^{13}$C-labeled TPP. Experiments were performed using a Bruker Avance III HD NMR spectrometer operating at a $^1$H frequency of 600 MHz and at a temperature of ~100 K. EmrE samples were prepared using a mixture of $^{13}$C$^{α,β}$-Tyr labeled EmrE (1.3 mg) mixed with unlabeled EmrE$^{E14Q}$ (1.3 mg), ~7 mg of deuterated DMPC (D$_{54}$), and a 2-fold molar excess of $^{13}$C-labeled TPP in 100 mM HEPES pH 6.2 in deuterated water. Proteoliposomes were centrifuged at 347,500 × $g$ using a Beckman Optima MAX-XP benchtop ultracentrifuge equipped with a TLA-110 rotor (Beckman Coulter). "DNP" juice was added to give a final concentration of 10 mM AMUPol in a 60/

30/10 (v/v/v) mixture of glycerol(D$_8$)/D$_2$O/H$_2$O. Two-dimensional $^{13}$C/$^{13}$C PDSD experiments were performed using a mixing time of 1 s and a MAS rate of 16 kHz in a 1.9 mm HCN low temperature HCN probe. A control was performed by adding TPP at natural abundance. No cross-peak was observed between EmrE and TPP for the control experiment.

All NMR spectra were processed in NMRPipe[65] version 8.1 and analyzed using Sparky[66] version 3.115.

## Synthesis of isotopically labeled tetraphenylphosphonium bromide

Tetraphenyl(phenyl-$^{13}$C$_6$)phosphonium bromide was synthesized via a palladium catalyzed coupling between triphenyl phosphine and $^{13}$C$_6$-bromobenzene as reported previously[67]. To a 10 ml rounded bottom flask was added $o$-xylene (1 ml), triphenyl phosphine (200 mg, 0.76 mmol), Pd$_2$(dba)$_3$ (7 mg, 1 mol %), followed by $^{13}$C$_6$-bromobenzene (80 μl, 0.76 mmol). The reaction was refluxed with stirring at 150 °C for 3 h, and the phosphonium salt began to precipitate out after 30 min. The salt was then washed with ether, dissolved in dichloromethane, and passed through a celite plug. Ether was again added to precipitate out the salt, and the supernatant was decanted. The solid was dissolved in water, filtered using a 0.2 μm filter, and the filtrate was lyophilized to give a pure white powder (115 mg, 43% yield). Perdeuterated tetraphenyl phosphonium (D$_{20}$) bromide was synthesized in an identical manner as described above, except starting from perdeuterated precursors of triphenyl phosphine (D$_{15}$, Sigma) and bromobenzene (D$_5$, Sigma). TPP derivatives were characterized by mass spectrometry and one-dimensional $^1$H and $^{13}$C NMR (Supplementary Fig. 10). $^{13}$C$_6$-TPP displayed an observed $m/z$ of 345.14 compared to an expected $m/z$ of 345.13; D$_{20}$-TPP displayed an observed $m/z$ of 359.25 compared to an expected $m/z$ of 359.13.

## Structure determination

Xplor-NIH (software version 2.52)[68] was used for the first stage of structure calculations. Ensembles of 256 structures were calculated using a simulated annealing protocol starting from extended structures: (1) proton-bound EmrE used EmrE$^{L51I}$ as monomer A and EmrE as monomer B and (2) TPP-bound EmrE used EmrE as monomer A and EmrE$^{E14Q}$ as monomer B. The initial temperature of 3500 K was cooled in steps of 2.5 K to a final temperature of 50 K. Initial torsion angle dynamics were performed for a total of 800 psec at 3500 K, followed by torsion angle dynamics for 1 psec at all other temperature steps. Additional steps of minimization in torsion angle and Cartesian angle space were performed. NMR restraints were implemented as flat-well potentials and described as follows. PRE distance restraints were calculated using intensity retentions similar to that previously described[69] and implemented to the amide hydrogen or methyl hydrogens from the spin label position imposed at the sulfur atom of cysteine, the CG1 atom of isoleucine, and the CG atom of leucine. For intensity retentions ($I_R$) ranging from 0.1 to 0.9, the target distance value used the Battiste and Wagner method[69]. Error ranges were ±5 Å with exceptions for the following intensity retention ranges: $I_R$ = 0 to 0.1 was imposed from 2.5 Å to 19.5 Å; $I_R$ = 0.1–0.3 was imposed with a lower bound of 2.5 Å; $I_R$ = 0.8–0.9 was imposed with no upper bound; and $I_R$ > 0.9 was imposed from 20 Å to no upper bound. NOE distance constraints were imposed with a range from 1.8 Å to 5.0 Å. MAS distance constraints from DARR or PDSD experiments at the indicated mixing times were implemented with the range in parentheses: 100 ms (2.5 Å to 5.5 Å), 250 ms (2.5 Å to 6.5 Å), 500 ms and longer (2.5 Å to 7.5 Å)[70]. Oriented sample constraints were implemented for the amide nitrogen in Xplor-NIH using the tensors, $\delta_{33}$ = 228.1 ppm, $\delta_{22}$ = 81.2 ppm, $\delta_{11}$ = 57.3 ppm[71], and a maximum dipolar coupling constant of 18.60 kHz, which was the largest dipolar splitting in uniformly $^{15}$N PISEMA spectra. The backbone error range implemented for chemical shift was ±1 ppm and dipolar coupling was ±0.25 kHz. Oriented sample constraints were

implemented for the indole nitrogen using the tensors, $\delta_{33} = 180.8$ ppm, $\delta_{22} = 129.6$ ppm, $\delta_{11} = 61$ ppm, and a maximum dipolar coupling constant of 17.62 kHz[72]. The tryptophan error range implemented for chemical shift was ±1.5 ppm and dipolar coupling was ±0.25 kHz. Of note, an observed DARR restraint from Glu14 in monomer A to Tyr60 in monomer B was imposed at a hydrogen bond distance from 1.5 Å to 4.5 Å and a strong $^{19}F$–$^{19}F$ intermonomer NOE between Trp63 residues was restrained from 1.8 Å to 4 Å. Finally, TALOS-N[73] dihedral angles were calculated using chemical shift assignments and imposed as restraints and hydrogen bond restraints were imposed for residues displaying helical dihedral angles from TALOS-N.

The second stage of structure calculations was refinement by taking the five lowest energy structures from the proton-bound and TPP-bound Xplor-NIH ensembles and performing all atom NMR-guided MD simulations. The structures were embedded into explicit lipid bilayers using CHARM-GUI[74,75]. Each system was setup following established models for membrane proteins including DMPC lipids[76], TIP3P[77] molecules, and sodium chloride; specific details of the system setup are provided in Supplementary Table 2. The CHARMM36 forcefield[78] was used with previously reported parameters for TPP[46] and implemented in GROMACS 2020.4. Hydrogen bonds to heavy atoms were constrained using the LINCs algorithm with a 2 fsec timestep[79]. Electrostatic interactions were treated with particle-mesh Ewald with 1.2 nm cutoff with a 0.12 nm grid[80]. The van der Waals interactions were switched off from 1 to 1.2 nm. A steepest descent minimization was completed until the maximum force was below 1000 kJ mol$^{-1}$ nm$^{-2}$. Next, the system was equilibrated for 10 nsec under constant volume temperature with heavy atoms of the protein constrained. From there, a constant pressure temperature simulation where heavy atoms of the protein (1000 kJ mol$^{-1}$ nm$^{-2}$) were slowly reduced to 0 kJ mol$^{-1}$ nm$^{-2}$ for a total of 30 nsec following an additional 30 nsec of unconstrained MD prior to production MD simulations. Temperature was maintained at 310.15 K using the Nose-Hoover thermostat[81]. Pressure was maintained at 1 bar using the Parrinello-Rahman coupling[82]. Structures were saved every 50 psec and subsequently used in MDAnalysis[83] and MDtraj[84] for analysis.

Selection of structures, i.e., NMR-guided MD simulations, were accomplished by calculating all NMR-derived distances and calculating the deviation from the experimental distances per structure. Violations were calculated using Eq. (1), where $d_i$ represents the calculated distance pair from MD simulation, $d_i^{exp}$ represents the experimental distance range associated with the restraint, and $n$ represents the total violations:

$$\text{Scaled Violation} = \sum_{i=1}^{n} \frac{|d_i - d^{exp}|}{1.5} \qquad (1)$$

Scaling the value was performed to remove any bias from large outliers. The three lowest violated structures were reseeded into the three replicates of MD simulations. This was performed in multiple cycles until the lowest violated structure did not improve after five rounds of reseeding. This yielded 17.8 μs and 13.3 μs of MD simulation for proton- and TPP-bound simulations, respectively. The lowest 10 scaled violated structures from all rounds of reseeding were used in final NMR-derived structures. The structures were then minimized using steepest descent minimization until the maximum force did not improve.

Due to the fluctuations in binding pocket in TPP-bound simulations, additional filtering of the structures was needed. These filtering steps selected structures that were consistent with NMR-derived distance restraints and the orientation of Trp63 side chains from oriented sample solid-state NMR. The latter was performed by determining chi1 and chi2 angles of the Trp63 side chain from the first round of 500 nsec of MD simulations that were consistent with the solid-state NMR dipolar couplings (|observed coupling – calculated coupling|<5 kHz) and anisotropic chemical shifts (|observed shift – calculated shift|<25

ppm). Chi1 and chi2 angles in agreement with solid-state NMR restraints resulted in the following cutoffs imposed in all subsequent rounds of reseeding. For Trp63 monomer A: (1) 155° < chi1 < −155° and −88° < chi2 < −35° or (2) −110° < chi1 < −60° and 50° < chi2 < 100°. For Trp63 monomer B: 155° < chi1 < −155° and 65° < chi2 < 115°. Only structures with no distance violations to TPP and correct Trp63 orientations were used for scaled violations calculations.

## Growth inhibition assays
Growth inhibition assays were performed in pET Duet-1 vectors with wild-type EmrE or single-site mutants[39]. Following transformation into *E. coli* BL21(DE3), a single colony was selected and grown for ~19 h at 37 °C in TBG medium supplemented with 20 μM IPTG and 100 μg/ml carbenicillin. Subsequently, cultures were diluted 300-fold into fresh TBG medium containing 20 μM IPTG, 100 μg/ml carbenicillin, and a range of ethidium bromide concentrations (0–500 μg/ml). The optical density at 600 nm (OD$_{600nm}$) was assessed every 15 min using a Bioscreen Pro C instrument at 37 °C with slow shaking. The OD$_{600nm}$ at 24 h was fitted to the non-linear function "*Sigmoidal, 4PL, X is concentration*" using Prism (software version 10.0.3 (217); GraphPad) to obtain the $IC_{50}$ value and the confidence interval at 95%. All resistance assays were independently performed at least two times, with a total of at least four replicates.

## Molecular dynamics simulations in DMPC lipid bilayers
Proton- and TPP-bound simulations were initiated from structures of the NMR-guided ensemble. Each structure was re-embedded into DMPC lipids and re-equilibrated following the same procedure as in the NMR-guided MD simulations. For proton-bound simulations, the L51I mutation in monomer A was mutated to leucine (the wild-type residue) and both Glu14 residues were protonated. For TPP-bound simulations, the E14Q mutation in monomer B was mutated to a protonated glutamic acid. MD simulations on deprotonated Glu14 of monomer A were initiated from the NMR-derived proton bound structure. MD simulations on the X-ray structure were initiated from PDB ID 7MH6 by protonating both Glu14 residues or protonating only Glu14 of monomer B and embedding in a DMPC lipid bilayer using the OPM server[85] for alignment and set up identical to NMR-derived structures in CHARMM-GUI[75]. All atom MD simulations were performed in triplicate where each replicate was run for 2.5 μs with three random velocities (three replicate simulations for each form of EmrE). Coordinates were saved every 50 psec and all structures were used in GROMACS 2020.4 and VMD[86] for the analysis. Cumulative MD simulation time yielded 7.5 μs which encompasses the timescales of side chain rotamer angle changes and local water dynamics measured in this study. Simulations were evaluated after 30 nsec of equilibration for the main text and Supplementary Figs. Notably, repeating the same analyses after 1 μs of equilibration displayed no significant changes relative to the 30 nsec equilibration (Supplementary Fig. 11).

## Reporting summary
Further information on research design is available in the Nature Portfolio Reporting Summary linked to this article.

# Data availability
The datasets generated during and/or analyzed during the current study are deposited in the Protein Data Bank and Biological Magnetic Resonance Bank. The proton-bound EmrE data are available under accession codes 8UWU and BMRB ID: 31125. TPP-bound EmrE data are available under accession codes 8UOZ and BMRB ID: 31121. MD simulation input and coordinates of starting and ending structure are publicly available using the link: https://doi.org/10.5281/zenodo.10999382. Published structures discussed or analyzed in this work: 7MH6, 7JK8, 7SFQ, 7SV9 and 3B5D. Source data are provided with this paper.

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

## Acknowledgements

This work was supported by NIH (R01AI108889, R01AI165782) and NSF awards (MCB1506420) to N.J.T. and NIH award (R35GM127040) to Y.Z. B.R.-C. acknowledges support from a CONACyT postdoctoral

fellowship. Solution NMR data collected from the 600 MHz spectrometer at NYU was supported with a cryoprobe purchase with funds from NIH (S10 OD016343). The solution NMR data collected from the 700 MHz spectrometer at the New York Structural Biology Center was made possible by a grant from ORIP/NIH facility improvement grant CO6RR015495. The 700 MHz spectrometer was purchased with funds from NIH grant S10OD018509. DNP solid-state NMR experiments were performed at the New York Structural Biology Center with assistance from Boris Itin. Structures and MD simulations utilized the HPC facilities of NYU. We thank Charles Schwieters for helpful suggestions about implementing the oriented sample restraints into Xplor-NIH.

## Author contributions

J.L. optimized and prepared NMR samples (solution NMR, MAS), acquired, processed, and analyzed NMR data, performed growth inhibition experiments, carried out and analyzed MD simulations, was involved in project design, and contributed to writing the manuscript. A.S. optimized and prepared the NMR samples (solution NMR, MAS, oriented), acquired, processed, and analyzed NMR data, acquired tryptophan fluorescence spectra, performed preliminary growth inhibition experiments, was involved in project design, and contributed to writing the manuscript. A.B. designed and performed NMR-guided MD simulations to determine the structural ensembles for proton-bound and TPP-bound EmrE and guided all production run MD simulations. B.R. optimized and prepared fluorinated EmrE, acquired, processed, and analyzed $^{19}$F NMR data, and performed some of the PRE experiments. M.C. optimized and prepared NMR samples (oriented) and acquired, processed, and analyzed oriented sample NMR data, and assigned solution NMR spectra of TPP-bound EmrE. B.R.-C. and M.C. contributed equally to this work. J.R.B. prepared preliminary NMR samples, acquired, processed, and analyzed NMR data, and was involved in optimizing Xplor-NIH scripts for structure calculation, C.M. performed EmrE ITC binding experiments, W.M.M. synthesized isotopically labeled TPP compounds, Y.Z. directed MD simulations, N.J.T. directed and designed the project and wrote the manuscript. All authors revised the manuscript.

## Competing interests

The authors declare no competing interests.
