## [Peer Review File · Nature Communications]

Dynamics Underlie the Drug Recognition Mechanism by the Efflux Transporter EmrEREVIEWER COMMENTS

Reviewer #1 (Remarks to the Author):

This manuscript by the Traaseth group uses a combination of NMR experiments and MD simulations to provide a structural basis for the drug binding to the membrane-embedded efflux pump EmrE. Findings are backed up by mutagenesis studies. This study is a true tour-de-force.

EmrE arranges as an unusual antiparallel dimer in the membrane that shuttles substrates using an alternated-access antiport mechanism. Upon deprotonation of a single Glu14 residues in the dimeric unit, EmrE changes from an occluded to an open conformation to which substrates can bind. In the open-conformation, binding affinity is increased by four orders of magnitude. Structures of EmrE at low pH (both Glu14 deprotonated) and high pH (one Glu protonated) were recently determined in a joint-effort by the groups of Mei Hong and Katherine Henzler-Wildman. The manuscript by Traaseth confirms most of the findings of Hong/ Henzler-Wildman, and rather completes some of the missing details. for Nature Communication. They show that the sidechain of the critical residue W63 is present in two rotameric conformations, one of which gets selected upon drug binding. While this is a subtle finding, given the interest in EmrE and technical prowess of the manuscript, I would suggest accepting this manuscript.

I only have a few comments:

- 1) In Figure 1A, the ^{13}C chemical shift of W63-E2 around 137 seems to virtually overlap with other aromatic signals, something which would hamper the analysis of restraints between Glu14 and W63 in the 2D CC ssNMR spectrum (Extended Figure 4). Can the authors please overlay the ^{13}C dimension of Figure 1A onto the 2D CC, and clearly show that they are able to identify the W63 signal?
- 2) In Figure 1A, state W63B (pH5/TPP) corresponds exactly to state W63A (pH5). What is the explanation for this?

Reviewer #2 (Remarks to the Author):

This manuscript provides more detail on how EmrE binds drugs and promotes their transport and is generally important as this is part of the major class of secondary active transport proteins known as small multidrug resistance (SMR) family. The authors mainly use NMR at various conditions to add to structural information regarding EmrE with/without a ligand, but nicely include molecular simulation work to support their findings and refine their NMR structures. Overall, I find this work worthy of publication in Nature Communications but some of my comments below should be addressed.

General Comments:

1. TPP-bound Structure: The authors present a structure of EmrE with TPP that places the small molecule in the upper half of the protein, i.e., near the lumen exit. The paper lacks any discussion on how this differs from past studies of structures with TPP-bound with a mutant (Nature Comm 13: p991) and nanobodies (Elife 11:e76766). These two studies appear to have TPP bound more at the center of EmrE. What structurally is different with these past studies compared to the current manuscript and why?

2. Computational Methods: The authors should properly reference force field parameters used in their work. They only reference the C36m force field for proteins and need to include the reference for lipids (J. Phys. Chem. B: 114: p7830). Moreover, there is no information given how TPP was introduced in these simulations as this is not a standard ligand in CHARMM. Did the authors use the CGenFF? If so, what were the penalty scores for the estimated parameters? If not, what force field was used.

Reviewer #3 (Remarks to the Author):

The research paper authored by Li and his team is a significant addition to the knowledge of the dynamics of Small Multidrug Transporter EmrE. The research reveals that the deprotonation of a single Glu14 residue in one monomer can shift the equilibrium towards the open state by changing its side-chain position and that of a nearby tryptophan residue. This structure change helps promote an open conformation that facilitates drug binding through a conformational selection mechanism. It also explains the well-documented increases in the binding affinity by approximately 2000-fold. However, the novelty of the rest of the manuscript needs to be clarified. It is unclear what, if any, the incremental contribution of the structures determined compared to the published NMR and X-ray structures.

This reviewer believes it could improve the paper's impact if the tone is changed. There's no need to overlook the existing literature to emphasize the significant and elegant contribution in understanding the dynamics of the transport process. Moreover, the paragraph that mentions, 'The current theory is that the structure of EmrE is essentially identical in the proton- and drug-bound states and therefore protons and drugs compete for binding in the substrate binding pocket' needs to be corrected to maintain the accuracy of the paper.

Since Jardetzki's proposal of an alternate access transport model, the axiom in the field has been that transport protein switches conformations to present the substrate binding site to alternate sides of the membrane without ever fully opening a channel from one side to the other. The alternating-access transport mechanism in these proteins ensures that the substrate is only accessible from one side of the membrane at any given time. This mechanism relies on complex and global protein conformational changes closely coupled to molecular events such as substrate binding and translocation. Although the authors' contribution is important and elegant, it is not unique. There are numerous examples of

transporters that change conformation upon proton or sodium binding. Therefore, the last paragraph of the Conclusion needs to be edited.

It's important to note that while non-specialists may use the terms "efflux pumps" and "efflux transporters" interchangeably, transporter experts must use more accurate and informative terms and emphasize the difference between the two. Pumps use redox, chemical, or light energy to move substrates uphill against their electrochemical potential. Ion-coupled transporters like EmrE, on the other hand, use the osmotic energy generated by primary pumps.

Specific comments:

The findings here seem to be in apparent contradiction with results reported by the Henzler-Wildman group, which concluded that simultaneous binding of TPP and protons is possible. This publication is not referenced here, and the contradiction is not discussed.

Very little is said about the second proton. Could the authors speculate when the second proton is released?

Minor comments:

Fig. 2b: the stoichiometry for transport is $2H^+$ /drug.

P.10, L. 3: TPP-bound EmrE crystals are at pH 7.25, and the pH 6.5 structure is with Me-TPP.

P.6, last line and more: Can you clarify the significance of this statement? In the published crystal structure without ligand (pH 5.2), EmrE possesses a deep, spacious aqueous pocket accessible from one side of the membrane.

P.7, last line and more: This statement wrongly assumes the omnipotence of NMR and MD data over crystal structures. It requires clarification on how they can reach such a conclusion.

Reviewer #4 (Remarks to the Author):

The paper by Li et al. describes a tour-de-force solid-state NMR study of the structure and dynamics of gating changes of the bacterial antibiotic drug efflux pump EmrE. The protein's structure has been previously determined by multiple techniques and under multiple ligand and gating conditions. EmrE is an inner E. coli inner membrane protein and forms a dimer of two equal subunits with 4 TM domains each. EmrE is a proton antiporter using the proton motive force as the energy source that clears drugs from cells by pumping them out. It is therefore of interest to understand the mechanism that couples proton influx to drug efflux – a problem to which this study offers important new insight.

While I cannot comment on the technical merits of the solid-state (SS) and some solution-state NMR approaches, the conclusions of the study seem clear and well documented: protonation of glutamate 14 in TM1 results in the flipping of the indole ring of tryptophan 63 in TM3 in a central region of the protein where these two residues are also in close proximity to each other and further interact with the same residues from the other subunit of the dimer. These ring flips open a cavity for binding of the model drug

tetraphenylphosphonium (TPP) as shown in the current paper. The authors support this mechanism by solving the structures of proton-bound and TPP-bound EmrE by SS NMR. The low pH proton-bound structure differs from another structure obtained by crystallography at low pH, in which the Trp 63 are not shown in a flipped conformation, which leads the authors to conclude that the crystal structure may not have been truly protonated. The conformational change of this side chain is supported by significant chemical shift changes of the indole nitrogen and adjacent carbon resonances in the SS NMR spectra upon protonation in low pH buffer compared to the unprotonated form at pH 9.

Overall, I believe this paper would benefit from a more extensive discussion of the implications of the results for the gating and drug transport mechanism. What do the results mean for the broader field? How does it change our view of how SMR transporters work? Why may this mechanism have been missed before? What other mechanisms have been proposed in the literature before and how common might this be across the large SMR family? To what extent is TPP representative of commonly used drugs and antibiotics? A cartoon of the mechanism may also help.

One thing that still puzzles me in this field is why the structures show the two subunits in an antiparallel fashion. Is this an artifact of the purification and sample preparation methods commonly employed (also in other papers on this protein) and how does a cell achieve this antiparallel insertion into the membrane (if it really does)? – Related to this, the authors show clearly that the two subunits do not change conformation simultaneously, but in an alternate fashion. That is easy to accept, but why could the drug not access the complex from both sides in symmetric biochemical samples like the ones used here and why are the two subunits that are distinguished by WT and the E14Q mutant not equivalent in drug binding from either side of the complex that is now asymmetric only by virtue of a very minor mutation in one subunit? This should be better explained.

Other points:

1. On line 4 of Results, Trp63 (and nearby Glu 14) are said to be in a central location. It might be helpful to refer the reader already at this point to Fig 2b which of course will only be discussed later in the text.
2. On p4/5, first reference to the inconsistency with the crystal structure of EmrE is made. Please state here the pH that was used for crystallization because this is important in this context.
3. Why were the PISEMA spectra of Fig 1b obtained at pH 5.8 with TPP and at pH 5.0 without TPP? Would it not have been better to compare the two at the same low pH?
4. MD simulations were performed starting from the protonated form of Glu 14 in the crystal structure and Trp 63 was seen to occasionally flip to the NMR structure under these conditions. An MD simulation starting from the unprotonated form potentially not showing a flip of the indole ring would be helpful to support the notion that Glu 14 in the crystal structure indeed may not have been protonated.

We thank the reviewers for their positive comments and suggestions for improving the manuscript. Below please find point-by-point responses in bold text.

Reviewer #1 comments

1) In Figure 1A, the ^{13}C chemical shift of W63-E2 around 137 seems to virtually overlap with other aromatic signals, something which would hamper the analysis of restraints between Glu14 and W63 in of the 2D CC ssNMR spectrum (Extended Figure 4). Can the authors please overlay the ^{13}C dimension of Figure 1A onto the 2D CC, and clearly show that they are able to identify the W63 signal?

In the revised manuscript, we juxtaposed the ^{13}C dimension from the $^{13}\text{C}/^{15}\text{N}$ correlation spectra onto the 2D ^{13}C - ^{13}C PDS from the same sample (see Extended Data Figure 4a). Notably, the Trp63-CE2 signal at approximately 139 ppm can be distinguished from other aromatic signals.

2) In Figure 1A, state W63B (pH5/TPP) corresponds exactly to state W63A (pH5). What is the explanation for this?

The ^{13}C and ^{15}N chemical shifts differ by ~ 0.7 ppm and ~ 0.2 ppm for the two peaks the reviewer mentions. For additional clarity on our chemical shift assignments of monomer A and B, we updated the manuscript to include $^{13}\text{C}/^{15}\text{N}$ correlation spectra obtained from conformationally biased samples (see Extended Data Figure 1e, f).

Nevertheless, there are two clusters of ^{15}N chemical shifts for Trp63 residues in the NMR spectra (~ 125 ppm and ~ 136 ppm). According to our structures, these two clusters differ in their chi2 rotamers for Trp63; in one cluster the indole NH faces away from the pocket and in the other it is oriented toward the pocket. The Trp63^B indole of TPP-bound EmrE at pH 5.0 retains the chi2 rotamer relative to the proton-bound state at pH 5.0. However, it experiences a chemical shift change as noted by the reviewer (i.e., it is closer to Trp63^A at pH 5.0). This relatively small perturbation is likely due to the reorientation of surrounding residues. In particular, Trp63^B is close to Glu14^B and this glutamate makes contacts with TPP which is likely to induce a small change in the environment for the NH indole of Trp63^B. Regardless, TPP binding does not alter the chi2 rotamer of Trp63^B according to our MAS spectra and structure and therefore remains near the ^{15}N chemical shift of ~ 125 ppm.

Reviewer #2 comments

1. TPP-bound Structure: The authors present a structure of EmrE with TPP that places the small molecule in the upper half of the protein, i.e., near the lumen exit. The paper lacks any discussion on how this differs from past studies of structures with TPP-bound with a mutant (Nature Comm 13: p991) and nanobodies (Elife 11:e76766). These two studies appear to have TPP bound more at the center of EmrE. What structurally is different with these past studies compared to the current manuscript and why?

In the revised manuscript, we expanded our original discussion regarding how TPP changes within the pocket relative to prior published structures of EmrE in complex with TPP (pages 10-11). This discussion is accompanied by overlays of our TPP-bound structure relative to the three previously reported TPP-bound structures (see Extended Data Figure 9a). The position of TPP in our structure differs from these structures by 2.5 Å to 7.6 Å, with the 2.5 Å difference corresponding to the NMR structure performed under similar conditions as our experiments. The 2.5 Å difference is small but significant and may arise from the larger number of experimental constraints we implemented for determining the structure compared to the prior work which primarily relied on TPP (fluorinated) contacts to EmrE and an X-ray C α model of EmrE as the starting structure (PDB ID 3B5D). It is also possible the fluorination of TPP may slightly alter the binding position. Larger differences in the TPP position relative to the prior NMR work at pH 8.0 and the X-ray crystal structure likely stems from a different protonation state of Glu14^B. The following text was added to the manuscript:

“Since EmrE binds TPP in the fully apo conformation and when only one of the two Glu14 residues is protonated³⁰, we compared our TPP-bound structure with those previously reported in complex with TPP. We found our structure

displays 3.4 Å and 3.6 Å backbone r.m.s.d. relative to NMR structures bound to fluorinated TPP at pH 5.8 (PDB ID 7JK8)²⁹ and pH 8.0 (PDB ID 7SFQ)²⁸, and a 2.0 Å backbone r.m.s.d. relative to an X-ray structure bound to TPP at pH 7.25 (PDB ID 7SV9)³¹ (Extended Data Fig. 9a). The TPP location in the substrate binding pocket differed on average by ~2.5 Å from the prior NMR structure at pH 5.8, ~4.2 Å from the prior NMR structure at pH 8.0, and ~7.6 Å from the X-ray structure. We also observed rotamer changes for Trp63 of monomer B with respect to each structure. The relatively high backbone r.m.s.d and deviations in the TPP location from the prior NMR structural work may arise from the reliance on a C α model crystal structure of EmrE (PDB ID 3B5D) to initiate the structure determination process and potential electrostatic differences between protonated and fluorinated TPP. Differences with the TPP binding position relative to the X-ray structure (PDB ID 7SV9), including a rotamer change for Trp63 of monomer B, may reflect a difference in Glu14 protonation within monomer B.

2. Computational Methods: The authors should properly reference force field parameters used in their work. They only reference the C36m force field for proteins and need to include the reference for lipids (J. Phys. Chem. B: 114: p7830). Moreover, there is no information given how TPP was introduced in these simulations as this is not a standard ligand in CHARMM. Did the authors use the CGenFF? If so, what were the penalty scores for the estimated parameters? If not, what force field was used.

TPP coordinates were introduced in MD simulations directly from our Xplor-NIH ensembles. TPP parameters utilized ones previously reported, where the authors parameterized TPP using CGenFF and a force field toolkit to optimize charges around the phosphorous center (<https://doi.org/10.1073/pnas.1722399115>). This article is cited in our manuscript, as well as references for lipids and water.

Reviewer #3 comments

The research paper authored by Li and his team is a significant addition to the knowledge of the dynamics of Small Multidrug Transporter EmrE. The research reveals that the deprotonation of a single Glu14 residue in one monomer can shift the equilibrium towards the open state by changing its side-chain position and that of a nearby tryptophan residue. This structure change helps promote an open conformation that facilitates drug binding through a conformational selection mechanism. It also explains the well-documented increases in the binding affinity by approximately 2000-fold. However, the novelty of the rest of the manuscript needs to be clarified. It is unclear what, if any, the incremental contribution of the structures determined compared to the published NMR and X-ray structures.

Our manuscript reports two significant and complementary findings: (1) characterization of conformational heterogeneity within the substrate binding pocket upon different proton and substrate bound conditions and (2) structural characterization of EmrE in well-defined protonation states. Namely, our structural experiments for proton-bound EmrE were performed at a pH value well below the pK_a values we previously reported for Glu14. In total, we collected 1,302 experimental restraints that were used to derive a structural ensemble of proton-bound EmrE in a partially occluded conformation, where aromatic residues, including Trp63, occupy the location of TPP in the substrate binding pocket. Oriented sample and MAS solid-state NMR played key roles in deciphering how the conformation of Trp63, located in the middle of the substrate binding pocket, changed upon protonation and drug binding (see Figure 1). Our proton-bound conformation is different than the X-ray structure (PDB ID 7MH6) in the accessibility of the substrate binding pocket which stems from the different orientation of the Trp63 residues. The X-ray structure contains a water molecule that crystallized in the central pocket and is likely filled with water, as reported by the authors. In contrast, our proton-bound structure excludes water, as inferred from the structural ensemble and seen in MD simulations. Hence, our structure represents a novel conformation of EmrE and one we propose to be the lowest energy conformation of the proton-bound state of EmrE.

We also determined an NMR structure of TPP-bound EmrE, where Gln14^B (mutant) was in the “proton-bound” conformation. Apart from the novelty of how the structure determination was performed, this structure also differs from the three prior reported TPP-bound structures of EmrE. Namely, the following is included in the Results section which highlights these differences:

“Since EmrE binds TPP in the fully apo conformation and when only one of the two Glu14 residues is protonated³⁰, we compared our TPP-bound structure with those previously reported in complex with TPP. We found our structure displays 3.4 Å and 3.6 Å backbone r.m.s.d. relative to NMR structures bound to fluorinated TPP at pH 5.8 (PDB ID 7JK8)²⁹ and pH 8.0 (PDB ID 7SFQ)²⁸, and a 2.0 Å backbone r.m.s.d. relative to an X-ray structure bound to TPP at pH 7.25 (PDB ID 7SV9)³¹ (Extended Data Fig. 9a). The TPP location in the substrate binding pocket differed on average by ~2.5 Å from the prior NMR structure at pH 5.8, ~4.2 Å from the prior NMR structure at pH 8.0, and ~7.6 Å from the X-ray structure. We also observed rotamer changes for Trp63 of monomer B with respect to each structure. The relatively high backbone r.m.s.d and deviations in the TPP location from the prior NMR structural work may arise from the reliance on a C α model crystal structure of EmrE (PDB ID 3B5D) to initiate the structure determination process and potential electrostatic differences between protonated and fluorinated TPP. Differences with the TPP binding position relative to the X-ray structure (PDB ID 7SV9), including a rotamer change for Trp63 of monomer B, may reflect a difference in Glu14 protonation within monomer B.”

Lastly, regarding the prior NMR structures, our TPP-bound structure represents a significant advancement, both in the number of experimental constraints and the fact it did not rely on prior X-ray structural work. Comparison of backbone r.m.s.d to these structures reveals an average value of ~3.5 Å, which is significantly different in the structural biology community.

Overall, our structures and measurements of conformational heterogeneity enabled us to unravel the molecular recognition mechanism of drug binding to EmrE. Without the combination of both sets of measurements, we would have been unable to derive this conclusion. The novelty of our findings is now more clearly articulated in the Discussion section.

This reviewer believes it could improve the paper's impact if the tone is changed. There's no need to overlook the existing literature to emphasize the significant and elegant contribution in understanding the dynamics of the transport process. Moreover, the paragraph that mentions, 'The current theory is that the structure of EmrE is essentially identical in the proton- and drug-bound states and therefore protons and drugs compete for binding in the substrate binding pocket' needs to be corrected to maintain the accuracy of the paper.

The resubmitted manuscript has been revised to adjust the tone. Furthermore, we added a paragraph to the Introduction section that better articulates contributions prior to ours in the EmrE field. This paragraph is displayed below with the last sentence being the modified one from the specific sentence quoted by the reviewer:

*“EmrE is a dual topology protein where the two monomers in the dimer are oppositely oriented in the inner membrane of *E. coli*^{22,23}. Substrates are moved across the membrane through a rocker-switch mechanism, resulting in accessibility of the substrate binding pocket to the cytoplasmic or periplasmic side of the membrane²⁴⁻²⁶. Covalent crosslinking of the dimer prevents alternating access and displays a loss of efflux activity²⁷, further underscoring the functional role of the anti-parallel structure. Recent atomic resolution models harmonize with the overall asymmetric and anti-parallel dimer quaternary structure of the earlier structural work and provide new insight into the transport cycle of EmrE. Namely, solid-state NMR spectroscopy in lipid bilayers was used to reveal how the high-affinity substrate tetraphenylphosphonium (TPP) changed positions in the substrate binding pocket as a function of protonation of Glu14 from one of the monomers^{28,29}. These findings offered a structural basis for observations showing that EmrE binds and transports TPP in the singly or doubly deprotonated states of Glu14³⁰. Likewise, EmrE crystal structures bound to different substrates revealed intermolecular contacts between EmrE and compounds varying in structure³¹. Notably, the similarity of the proton-bound and TPP-bound structures (0.376 Å backbone r.m.s.d.) suggests drug binding occurs predominantly through the competition model³², which postulates protons and drugs compete for binding to Glu14 residues in the substrate binding pocket.”*

Furthermore, we underscore in the Discussion section how the X-ray structures are highly complementary to our manuscript. Namely, by using NMR spectroscopy and MD simulations, we were able to offer a more complete understanding of drug binding and the role of Glu14 protonation (see the sentence below from the Discussion section).

“Hence, analyses of NMR spectroscopy and MD simulations in this work together with X-ray crystallographic findings³¹ enabled a more complete structural basis of drug binding (Fig. 5).”

Since Jardetzki's proposal of an alternate access transport model, the axiom in the field has been that transport protein switches conformations to present the substrate binding site to alternate sides of the membrane without ever fully opening a channel from one side to the other. The alternating-access transport mechanism in these proteins ensures that the substrate is only accessible from one side of the membrane at any given time. This mechanism relies on complex and global protein conformational changes closely coupled to molecular events such as substrate binding and translocation. Although the authors' contribution is important and elegant, it is not unique. There are numerous examples of transporters that change conformation upon proton or sodium binding. Therefore, the last paragraph of the Conclusion needs to be edited.

In our revised manuscript, we replaced the Conclusion section with a Discussion section and expanded its content. Here, we discuss other mechanisms reported in the transporter field, including gate-like models and antiport mechanisms. The Discussion section is displayed below for convenience:

“Four of the five multidrug efflux families carry out secondary active transport by harnessing the PMF or differences in solute concentrations across the membrane. Antiporters like EmrE rely on the alternating access mechanism to transport substrates and protons by switching between inward-facing and outward-facing directions. In fact, 12-TM domain transporters from the Major Facilitator Superfamily (MFS) and others occupy additional conformations along the transport cycle, including inward-open, inward-occluded, occluded, outward-occluded, and outward-open states^{16,48,49}. Switch-like mechanisms have been proposed for MFS family members where protonation of membrane embedded aspartate and glutamate residues regulate opening and closing of the substrate binding pocket. Recent findings on QacA⁵⁰, MdfA⁵¹, and NorA⁵² describe how protonation modulates conformational changes between such inward- and outward-facing states. Measurements sensitive to dynamics, such as single molecule fluorescence experiments, NMR spectroscopy, and EPR spectroscopy, have complemented structural findings by showing how substrate and ion binding modulate the rate of conformational switching between inward- and outward-facing conformations^{33,53,54}.

The novelty of this work is the simultaneous characterization of structure and conformational heterogeneity under well-defined Glu14 protonation states of EmrE. Our findings reveal how deprotonation of the membrane embedded Glu14 residue in monomer A disrupts a hydrogen bond between its carboxyl group and the backbone carbonyl of Trp63 in TM3 (Fig. 5). Disruption of this interaction induces the side chain of Trp63 in monomer A to populate two states that resemble the occluded proton-bound conformation and the open TPP-bound conformation. Substrates bind to the latter conformation, indicating a molecular recognition mechanism involving conformational selection. Hence, acid/base chemistry at Glu14 of monomer A modulates an equilibrium change toward the open state of the transporter. This model provides an explanation to the pH dependence of drug binding (Extended Data Fig. 1a). Such an experimental structure of an occluded conformation we determined for the proton-bound state has not previously been reported for EmrE or another SMR family transporter. However, the presence of an occluded conformation for EmrE was proposed based on distance measurements from EPR spectroscopy⁵⁵ and MD simulations⁴⁶. We hypothesize our structure is significant since it likely serves as an intermediate conformation between outward-open and inward-open conformations, resulting in the movement of protons from the periplasm to the cytoplasm. This interpretation is consistent with faster alternating access exchange for proton-bound EmrE compared to deprotonated states of EmrE at higher pH values³³.

What additional structural evidence supports the conformational selection model? Guided by NMR measurements directly probing Trp63 within the substrate binding pocket, we propose that the drug-free X-ray structure of EmrE³¹ likely corresponds to one of the conformations EmrE samples when Glu14 of monomer A becomes deprotonated. Indeed, the presence of a water molecule in the substrate binding pocket of this crystal structure is consistent with MD simulations showing that deprotonation of Glu14 of monomer A leads to greater water penetration into the pocket. Hence, analyses of NMR spectroscopy and

MD simulations in this work together with X-ray crystallographic findings³¹ enabled a more complete structural basis of drug binding (Fig. 5).

In conclusion, our results provide a model for how dynamics are modulated upon deprotonation of a single glutamate residue within the EmrE dimer. The corollary is that acid/base chemistry of membrane embedded acidic residues serves as a trigger for equilibrium changes, not conformational switching reported in other efflux systems. We propose that substrates bind to the open-like conformation of the equilibrium through conformational selection, ultimately resulting in a shift toward the substrate-bound state through Le Chatelier's principle. The presence of glutamate and aspartate residues commonly found in other proton-coupled transporters suggests ionization of membrane embedded acidic residues could be a common mechanism for modulating equilibrium shifts and the accessibility of the substrate binding pocket.”

It's important to note that while non-specialists may use the terms "efflux pumps" and "efflux transporters" interchangeably, transporter experts must use more accurate and informative terms and emphasize the difference between the two. Pumps use redox, chemical, or light energy to move substrates uphill against their electrochemical potential. Ion-coupled transporters like EmrE, on the other hand, use the osmotic energy generated by primary pumps.

In the revised document, we characterized EmrE as an efflux transporter, to avoid confusion with primary active transporters. This nomenclature is consistent with EmrE's role as a secondary active transporter that harnesses the proton motive force resulting in drug efflux.

The findings here seem to be in apparent contradiction with results reported by the Henzler-Wildman group, which concluded that simultaneous binding of TPP and protons is possible. This publication is not referenced here, and the contradiction is not discussed.

The TPP-bound structure we report in our manuscript corresponds to EmrE bound with one proton and one TPP. In fact, this was the reason we used EmrE/EmrE^{E14Q} heterodimers to derive several constraints (i.e., E14Q serves as a mimic of a protonated Glu14). Hence, our findings agree with the Henzler-Wildman article (<https://doi.org/10.1073/pnas.1708671114>), which is cited in the manuscript (reference #30).

Very little is said about the second proton. Could the authors speculate when the second proton is released?

The second proton is released upon drug binding to the cytoplasmic facing conformation of EmrE. We reported the p*K_a* value for Glu14^B of 8.4 in the proton-bound state (<https://doi.org/10.1073/pnas.2110790118>) and the Henzler-Wildman group reported the p*K_a* value of 6.8 of this glutamate when bound to TPP (<https://doi.org/10.1073/pnas.1708671114>). Hence, for a cytoplasmic pH value of approximately 7.5 (common in bacteria like *E. coli*), the singly proton-bound state is favored in the absence of drug, while the fully deprotonated form is favored after TPP binding. We reported a detailed transport cycle in our previous publication (see Figure 7 in <https://doi.org/10.1073/pnas.2110790118>). A schematic for the release of the second proton is also shown in Figure 5 of the current manuscript, which highlights the same mechanism.

Minor comments:

Fig. 2b: the stoichiometry for transport is 2H⁺/drug.

Figure 2b was adjusted to 2H⁺.

P.10, L. 3: TPP-bound EmrE crystals are at pH 7.25, and the pH 6.5 structure is with Me-TPP.

Thank you for pointing out this mistake. We updated the pH to 7.25 in the revised manuscript.

P.6, last line and more: Can you clarify the significance of this statement? In the published crystal structure without ligand (pH 5.2), EmrE possesses a deep, spacious aqueous pocket accessible from one side of the membrane.

Indeed, this quoted text from Stockbridge and co-workers (<https://doi.org/10.7554/eLife.76766>) underscores a key point of our manuscript. Namely, in our proton-bound structure of EmrE, we observed a solvent-excluded and occluded state while the X-ray structure shows “a deep, spacious aqueous pocket” (<https://doi.org/10.7554/eLife.76766>). Together with MD simulations, this is the basis for our sentence on page

13 of the manuscript: “Based on these findings, it is likely the drug-free X-ray structure corresponds to Glu14 in a deprotonated state.” Our interpretation is also consistent with ITC derived K_d values showing EmrE with lower affinity to TPP at acidic pH values and higher affinity at basic pH values (see Extended Data Fig. 1a).

P.7, last line and more: This statement wrongly assumes the omnipotence of NMR and MD data over crystal structures. It requires clarification on how they can reach such a conclusion.

The statement the reviewer refers to in the original manuscript was: “Based on these findings and the inconsistency of Trp63 orientations with PISEMA experiments, we conclude that the X-ray structure does not reflect the low energy proton-bound conformation.” The goal of this sentence was not to convey omnipotence of NMR and MD over crystal structures. We are an interdisciplinary group that uses several biophysical techniques. Rather, our manuscript draws the following conclusion which is supported by NMR data and MD simulations: “the drug-free X-ray structure corresponds to Glu14 in a deprotonated state.” This conclusion was made after determining the proton-bound structure of EmrE, which involved the collection of 1,302 experimental constraints. Based on these restraints (e.g., the oriented sample data directly probing the side chain of Trp63) and MD simulations performed on the NMR structure and the X-ray structure in protonated and deprotonated states, we are confident in the conclusion that our structure represents the most favorable conformation of proton-bound EmrE. As such, we adjusted the sentence referred to by the reviewer as follows:

“These simulation results support the conclusion that our NMR-derived structure represents the lowest energy conformation of proton-bound EmrE.”

Reviewer #4 comments

Overall, I believe this paper would benefit from a more extensive discussion of the implications of the results for the gating and drug transport mechanism. What do the results mean for the broader field? How does it change our view of how SMR transporters work? Why may this mechanism have been missed before? What other mechanisms have been proposed in the literature before and how common might this be across the large SMR family? To what extent is TPP representative of commonly used drugs and antibiotics? A cartoon of the mechanism may also help.

In the revised manuscript, we included an expanded Discussion section (see quoted text below) that addresses several of the questions posed by the reviewer and a cartoon schematic (Figure 5) to better articulate the mechanism proposed in this manuscript. As we allude to below, it is likely that our measurements directly on Trp63 (centrally located in the substrate binding pocket) in lipid bilayers under well-defined protonation states of Glu14 provided key clues into the mechanism. Since these are the first measurements of their kind reported for EmrE, this may be a reason the observation was not previously reported. In terms of the nature of TPP as a typical antibiotic transported by the SMR family, additional measurements will be needed to determine whether EmrE and other transporters from the family utilize the same two position TPP binding mechanism displayed in Figure 5. While TPP shares an aromatic character of other substrates transported by EmrE, it is a bit more globular than some of the more planar substrates commonly transported by efflux systems. It is likely that complementary usage of NMR spectroscopy, MD simulations, and X-ray crystallography will provide a valuable approach for understanding whether this mechanism is shared in other substrates.

“Four of the five multidrug efflux families carry out secondary active transport by harnessing the PMF or differences in solute concentrations across the membrane. Antiporters like EmrE rely on the alternating access mechanism to transport substrates and protons by switching between inward-facing and outward-facing directions. In fact, 12-TM domain transporters from the Major Facilitator Superfamily (MFS) and others occupy additional conformations along the transport cycle, including inward-open, inward-occluded, occluded, outward-occluded, and outward-open states^{16,48,49}. Switch-like mechanisms have been proposed for MFS family members where protonation of membrane embedded aspartate and glutamate residues regulate opening and closing of the substrate binding pocket. Recent findings on QacA⁵⁰, MdfA⁵¹, and NorA⁵² describe how protonation modulates conformational changes between such inward- and outward-facing states. Measurements sensitive to dynamics, such as single molecule fluorescence

experiments, NMR spectroscopy, and EPR spectroscopy, have complemented structural findings by showing how substrate and ion binding modulate the rate of conformational switching between inward- and outward-facing conformations^{33,53,54}.

The novelty of this work is the simultaneous characterization of structure and conformational heterogeneity under well-defined Glu14 protonation states of EmrE. Our findings reveal how deprotonation of the membrane embedded Glu14 residue in monomer A disrupts a hydrogen bond between its carboxyl group and the backbone carbonyl of Trp63 in TM3 (Fig. 5). Disruption of this interaction induces the side chain of Trp63 in monomer A to populate two states that resemble the occluded proton-bound conformation and the open TPP-bound conformation. Substrates bind to the latter conformation, indicating a molecular recognition mechanism involving conformational selection. Hence, acid/base chemistry at Glu14 of monomer A modulates an equilibrium change toward the open state of the transporter. This model provides an explanation to the pH dependence of drug binding (Extended Data Fig. 1a). Such an experimental structure of an occluded conformation we determined for the proton-bound state has not previously been reported for EmrE or another SMR family transporter. However, the presence of an occluded conformation for EmrE was proposed based on distance measurements from EPR spectroscopy⁵⁵ and MD simulations⁴⁶. We hypothesize our structure is significant since it likely serves as an intermediate conformation between outward-open and inward-open conformations, resulting in the movement of protons from the periplasm to the cytoplasm. This interpretation is consistent with faster alternating access exchange for proton-bound EmrE compared to deprotonated states of EmrE at higher pH values³³.

What additional structural evidence supports the conformational selection model? Guided by NMR measurements directly probing Trp63 within the substrate binding pocket, we propose that the drug-free X-ray structure of EmrE³¹ likely corresponds to one of the conformations EmrE samples when Glu14 of monomer A becomes deprotonated. Indeed, the presence of a water molecule in the substrate binding pocket of this crystal structure is consistent with MD simulations showing that deprotonation of Glu14 of monomer A leads to greater water penetration into the pocket. Hence, analyses of NMR spectroscopy and MD simulations in this work together with X-ray crystallographic findings³¹ enabled a more complete structural basis of drug binding (Fig. 5).

In conclusion, our results provide a model for how dynamics are modulated upon deprotonation of a single glutamate residue within the EmrE dimer. The corollary is that acid/base chemistry of membrane embedded acidic residues serves as a trigger for equilibrium changes, not conformational switching reported in other efflux systems. We propose that substrates bind to the open-like conformation of the equilibrium through conformational selection, ultimately resulting in a shift toward the substrate-bound state through Le Chatelier's principle. The presence of glutamate and aspartate residues commonly found in other proton-coupled transporters suggests ionization of membrane embedded acidic residues could be a common mechanism for modulating equilibrium shifts and the accessibility of the substrate binding pocket.”

One thing that still puzzles me in this field is why the structures show the two subunits in an antiparallel fashion. Is this an artifact of the purification and sample preparation methods commonly employed (also in other papers on this protein) and how does a cell achieve this antiparallel insertion into the membrane (if it really does)? – Related to this, the authors show clearly that the two subunits do not change conformation simultaneously, but in an alternate fashion. That is easy to accept, but why could the drug not access the complex from both sides in symmetric biochemical samples like the ones used here and why are the two subunits that are distinguished by WT and the E14Q mutant not equivalent in drug binding from either side of the complex that is now asymmetric only by virtue of a very minor mutation in one subunit? This should be better explained.

We expanded our Introduction section to better explain the well-established dual topology feature of EmrE. Below is the portion of the paragraph from the manuscript that addresses the reviewer's questions regarding the antiparallel nature of EmrE.

“EmrE is a dual topology protein where the two monomers in the dimer are oppositely oriented in the inner membrane of E. coli^{22,23}. Substrates are moved across the membrane through a rocker-switch mechanism, resulting in accessibility of the substrate binding pocket to the cytoplasmic or periplasmic side

of the membrane²⁴⁻²⁶. Covalent crosslinking of the dimer prevents alternating access and displays a loss of efflux activity²⁷, further underscoring the functional role of the anti-parallel structure. Recent atomic resolution models harmonize with the overall asymmetric and anti-parallel dimer quaternary structure of the earlier structural work and provide new insight into the transport cycle of EmrE.”

Reference #22 describes how EmrE variants (EmrEⁱⁿ or EmrE^{out}) that insert into the membrane in a unidirectional manner are unable to confer resistance when expressed alone. However, co-expression of both variants rescues the phenotype observed for wild-type EmrE. These experiments, together with the cited structural studies, established the basis for the dual topological nature of EmrE, which ultimately stems from a lack of positive inside rule (explained in reference #22).

Regarding the accessibility of the pocket, as indicated in the quoted paragraph from the Introduction (see above), EmrE is open to one side of the membrane at a time – either the cytoplasmic or periplasmic side but not both. This is consistent with the alternating access model. The EmrE/EmrE^{E14Q} heterodimer when bound to TPP was valuable in our experiments since it biased the conformation of the EmrE monomer to display “A” peaks and the E14Q mutant monomer to display “B” peaks. Although this conformation can be thought of as cytoplasmic or periplasmic facing, it has the advantage of suppressing conformational exchange between monomers A and B which simplifies NMR data interpretation. We included a schematic of the conformational bias effect in Figure 3a to illustrate this point more clearly.

Other points:

1. On line 4 of Results, Trp63 (and nearby Glu 14) are said to be in a central location. It might be helpful to refer the reader already at this point to Fig 2b which of course will only be discussed later in the text.

We added a panel in Extended Data Figure 1d to guide the reader to the location of Trp63 in EmrE.

2. On p4/5, first reference to the inconsistency with the crystal structure of EmrE is made. Please state here the pH that was used for crystallization because this is important in this context.

We updated this sentence to: “Overall, the large perturbation observed for Trp63 of monomer A was not explained by X-ray crystal structures of EmrE (drug-free at pH 5.2; TPP-bound at pH 7.25), which displayed essentially the same structure bound or unbound to TPP³¹”.

3. Why were the PISEMA spectra of Fig 1b obtained at pH 5.8 with TPP and at pH 5.0 without TPP? Would it not have been better to compare the two at the same low pH?

Ideally, performing all measurements at the same pH would have been preferable. However, we selected a more acidic pH value for experiments in the absence of TPP due to a slightly improved overall spectra quality. For PISEMA experiments on TPP-bound EmrE, we used a slightly higher pH value due to the poorer binding affinity as the pH is lowered (e.g., see Extended Data Fig. 1a). Regardless of this small pH difference, the samples are under well-defined protonation states of Glu14.

4. MD simulations were performed starting from the protonated form of Glu 14 in the crystal structure and Trp 63 was seen to occasionally flip to the NMR structure under these conditions. An MD simulation starting from the unprotonated form potentially not showing a flip of the indole ring would be helpful to support the notion that Glu 14 in the crystal structure indeed may not have been protonated.

In our original submission, we reported that deprotonation of Glu14 induces conformational heterogeneity of the protein at Trp63, as observed in MAS experiments (Figure 1a, middle) and in MD simulations (Figure 4d). Thus, our hypothesis is that the crystal structure deprotonated at Glu14 would also experience Trp63 flipping.

In the revised manuscript, we performed the simulations suggested by the reviewer (i.e., initiating simulations from the X-ray structure after deprotonating Glu14^A). These new results are in striking agreement with similar simulations on deprotonated Glu14^A initiated from our NMR structure (Extended Data Figure 7c-f). Namely, in one replicate, Trp63^A flipped its indole orientation after ~1.6 μ sec and remained constant for an additional ~0.9 μ sec (Extended Data Fig. 7d, f). Note that the observed indole ring flipping differs slightly from the flipping observed in simulations initiated from the protonated state of the crystal structure. Specifically, in simulations where Glu14^A is deprotonated, we did not observe interactions between the backbone carbonyl oxygen of

Trp63^A and the side chain carboxyl oxygens of Glu14^A. In contrast, such interactions were seen in one replicate of the proton-bound EmrE simulations. Hence, the MD simulations support the conformational selection model proposed in our manuscript, which concludes that EmrE experiences conformational heterogeneity at Trp63^A upon Glu14^A deprotonation.

REVIEWERS' COMMENTS

Reviewer #1 (Remarks to the Author):

I am satisfied with the changes made by the authors. I like to congratulate the authors, this is a very strong study that looks even more exciting in the revised version.

Reviewer #2 (Remarks to the Author):

updated manuscript is acceptable for publication.

Reviewer #3 (Remarks to the Author):

The authors have satisfactorily addressed my comments.

Reviewer #4 (Remarks to the Author):

The authors have answered all my questions and have significantly improved the manuscript by putting their results in better context with previous studies and added a new figure explaining the how the new structures add to explain the mechanism of substrate transport by EmrE. I support publication of the manuscript as presented.